# Composition Theorems for Interactive Differential Privacy

**Xin Lyu**
Department of Electrical Engineering and Computer Science
University of California, Berkeley
Berkeley, CA, 94720
xinlyu@berkeley.edu

## Abstract

An interactive mechanism is an algorithm that stores a data set and answers adaptively chosen queries to it. The mechanism is called differentially private, if any adversary cannot distinguish whether a specific individual is in the data set by interacting with the mechanism. We study composition properties of differential privacy in concurrent compositions. In this setting, an adversary interacts with $k$ interactive mechanisms in parallel and can interleave its queries to the mechanisms arbitrarily. Previously, Vadhan and Wang [2021] proved an optimal concurrent composition theorem for pure-differential privacy. We significantly generalize and extend their results. Namely, we prove optimal parallel composition properties for several major notions of differential privacy in the literature, including approximate DP, Rényi DP, and zero-concentrated DP. Our results demonstrate that the adversary gains no advantage by interleaving its queries to independently running mechanisms. Hence, interactivity is a feature that differential privacy grants us for free.

Concurrently and independently of our work, Vadhan and Zhang [2022] proved an optimal concurrent composition theorem for $f$-DP [Dong et al., 2022], which implies our result for the approximate DP case.

## 1 Introduction

By now, differential privacy [Dwork et al., 2006b] has been widely accepted as a standard framework for protecting individual privacy when performing data analysis on data sets that may contain sensitive information of individuals (see, e.g., the surveys by Dwork and Roth [2014], Vadhan [2017]).

Let $\mathcal{M}$ be an algorithm that runs on a data set $x$ and calculates some information about it. Roughly speaking, $\mathcal{M}$ is called differentially private, if the output distribution of $\mathcal{A}$ remains nearly identical when we arbitrarily modify a single entry in $x$.

One essential feature of differential privacy is its *composability*. Composition captures the scenario where a data analyst runs $k$ differentially private algorithms sequentially, and releases the results afterward. Typically, a composition theorem has the following form: if each of the $k$ algorithms satisfies differential privacy, then the analyst's output is still differentially private with moderately degraded privacy parameters.

Composition theorems are important for at least two reasons. First, we might want to perform computation tasks on the same data set multiple times and still have reasonable control over the privacy loss. In this case, composition theorems reveal how the privacy guarantee degrades over time. More importantly, composition theorems allow us to build more complex and powerful differentially-

36th Conference on Neural Information Processing Systems (NeurIPS 2022).

private algorithms from simple primitives, and argue the privacy guarantee of the combined algorithm in a straightforward way.

There is a rich literature concerning the composition property of differential privacy (see, e.g., Dwork et al. [2006a, 2010], Kairouz et al. [2015], Murtagh and Vadhan [2018], Bassily et al. [2021]). However, most existing composition theorems only consider the scenario where an analyst runs several private algorithms *sequentially*. That is, the analyst will only move on to the next algorithm after finishing their computation with the previous one. In contrast, many fundamental primitives in differential privacy are interactive in nature, such as the sparse vector technique [Dwork et al., 2009, Roth and Roughgarden, 2010] and private multiplicative weight updates [Hardt and Rothblum, 2010]. Hence, the interactivity issue appears to be a significant limitation of current composition theorems. Namely, the data analyst may want to communicate with several interactive mechanisms *concurrently*, and interleave its queries to the mechanisms arbitrarily. A sequential composition theorem completely fails to capture this scenario. Also, in practice, deployments of DP algorithms often demand a better understanding of concurrent compositions of interactive mechanisms [Hay et al., 2020].

Recently, Vadhan and Wang [2021] initiated a study of concurrent compositions and proved an optimal concurrent composition theorem for pure differential privacy. In this work, we significantly advance this research direction by proving optimal concurrent composition theorems for several popular notions of differential privacy, including approximate DP, Rényi DP, zero-concentrated DP and truncated concentrated DP.

## 1.1 Setup

Before we continue, we set up necessary pieces of notation. We use $\mathcal{X}$ and $\mathcal{Y}$ to denote the domain of query messages and responses, respectively. We assume that both $\mathcal{X}$ and $\mathcal{Y}$ are finite sets. This assumption is for easing some mathematical manipulation and is not restrictive: all practical applications of differential privacy have finite input and output spaces anyway.

For a set $S$, denote by $\Delta(S)$ the set of all possible distributions supported over $S$. We define interactive systems below.

**Definition 1** (Interactive system). An interactive system is a (randomized) algorithm $\mathcal{M} \colon (\mathcal{X} \times \mathcal{Y})^* \times \mathcal{X} \to \Delta(\mathcal{Y})$. The input to $\mathcal{M}$ is an interaction history $(x_1, y_1), (x_2, y_2), \ldots, (x_t, y_t) \in (\mathcal{X} \times \mathcal{Y})^t$ together with a query $x_{t+1}$. The output of $\mathcal{M}$ is denoted by $y_{t+1} \sim \mathcal{M}((x_i, y_i)_{i \in [t]}, x_{t+1})$.

A technicality worth mentioning is that due to the internal memory and randomness of an interactive system $\mathcal{M}$, the response of $\mathcal{M}$ to the $(t + 1)$-th query might be correlated with its responses to previous queries. Although the internal randomness of $\mathcal{M}$ is not explicitly stated as a parameter, Definition 1 captures this correlation by requiring that each query $x_{t+1}$ to $\mathcal{M}$ is attached with the interaction history $(x_1, y_1), \ldots, (x_t, y_t)$. This history is sufficient for determining the conditional distribution of the response $\mathcal{M}((x_i, y_i)_{i \in [t]}, x_{t+1})$ without specifying the internal randomness and memory.

We make a distinction between mechanisms and systems. By "mechanism" we mean a differentially private algorithm $\mathcal{M}$ that holds a sensitive input $d$ and answers queries about it. When applied to a concrete input $d$, $\mathcal{M}$ induces an interactive system, denoted by $\mathcal{M}^d$. According to the definition of differential privacy, studying the privacy of a mechanism boils down to studying the pair of systems $(\mathcal{M}^d, \mathcal{M}^{d'})$ induced by running $\mathcal{M}$ on every pair of neighboring inputs $(d, d')$. For brevity, we usually assume W.L.O.G. that the input only consists of a single bit $b \in \{0, 1\}$, and we compare the two systems $\mathcal{M}^0, \mathcal{M}^1$ induced by $\mathcal{M}$.

**Concurrent composition.** We define concurrent composition of interactive systems. Suppose $\mathcal{M}_1, \mathcal{M}_2, \ldots, \mathcal{M}_k$ are $k$ systems. The concurrent composition of them is an interactive system $\mathrm{COMP}(\mathcal{M}_1 \ldots \mathcal{M}_k)$ with query domain $[k] \times \mathcal{X}$ and response domain $\mathcal{Y}$. An adversary is a (possibly randomized) query algorithm $\mathcal{A} \colon ([k] \times \mathcal{X} \times \mathcal{Y})^* \to \Delta([k] \times \mathcal{X})$. The interaction between $\mathcal{A}$ and $\mathrm{COMP}(\mathcal{M}_i)$ is a stochastic process that runs as follows. $\mathcal{A}$ first[1] computes a pair $(i_1, x_1) \in [k] \times \mathcal{X}$, sends a query $x_1$ to $\mathcal{M}_{i_1}$ and gets the response $y_1$. In the $t$-th step, $\mathcal{A}$ calculates the next pair $(i_t, x_t)$ based on the history, sends the $t$-th query $x_t$ to $\mathcal{M}_{i_t}$ and receives $y_t$. There is no communication or

---

[1]We assume it is always the adversary who sends the first message. This is without loss of generality: we can let the first message sent from the adversary to each system be an "Initiliazation" query. Having received the initialization query, the system returns either a starting message or simply a "SUCCESS" symbol.

interaction between the interactive systems. Each system $\mathcal{M}_i$ can only see its own interaction with $\mathcal{A}$. Let $\mathbf{IT}(\mathcal{A} : \mathcal{M}_1, \ldots, \mathcal{M}_k)$ denote the random variable recording the transcript of the interaction.

In the special case $k = 1$, there is only one system $\mathcal{M}$ and the adversary is interacting with it. We define approximate differential privacy for interactive mechanisms in this case.

**Definition 2** (Indistinguishability and $(\varepsilon, \delta)$-DP)**.** Two interactive systems $\mathcal{M}^0, \mathcal{M}^1$ are called $(\varepsilon, \delta)$-indistinguishable, if for every $b \in \{0, 1\}$, every adversary $\mathcal{A}$ and every collection of transcripts $S \subseteq \{(x_i, y_i)_{i \in [T]}\}$, it holds that

$$\Pr[\mathbf{IT}(\mathcal{A} : \mathcal{M}^b) \in S] \leq e^\varepsilon \Pr[\mathbf{IT}(\mathcal{A} : \mathcal{M}^{1-b}) \in S] + \delta. \tag{1}$$

Let $\mathcal{M}$ be an interactive mechanism. $\mathcal{M}$ is called $(\varepsilon, \delta)$-approximate differentially private (or $(\varepsilon, \delta)$-DP for short), if for every two neighboring data sets $d$ and $d'$, the systems $\mathcal{M}^d$ and $\mathcal{M}^{d'}$ are $(\varepsilon, \delta)$-indistinguishable.

## 1.2 Differential Privacy in Concurrent Compositions

We study the privacy guarantee under concurrent compositions. Let $\mathcal{M}_1^b, \ldots, \mathcal{M}_k^b$ be $k$ interactive mechanisms, each satisfying $(\varepsilon, \delta)$-DP. Consider their concurrent composition $\mathrm{COMP}(\mathcal{M}_1^b \ldots \mathcal{M}_k^b)$. We want to find out the smallest parameters $\varepsilon', \delta'$ such that $\mathrm{COMP}(\mathcal{M}_i^b)$ satisfies $(\varepsilon', \delta')$-DP. In the sequential composition, the adversary $\mathcal{A}$ interacts with $\mathcal{M}_i$'s in order and cannot interleave its queries. In this case, it is known by the advanced composition theorem [Dwork et al., 2010] that $\mathbf{IT}(\mathcal{A} : \mathcal{M}_1^0, \ldots, \mathcal{M}_k^0)$ and $\mathbf{IT}(\mathcal{A} : \mathcal{M}_1^1, \ldots, \mathcal{M}_k^1)$ are $(O(\sqrt{k \log(1/\delta')}\varepsilon), k\delta + \delta')$-indistinguishable.

However, in general, the adversary can interleave its queries arbitrarily, and the differential privacy guarantee warranted by $\mathrm{COMP}(\mathcal{M}_i)$ is less clear. Vadhan and Wang [2021] were the first to formally study this question. They showed that in the special case $\delta = 0$, an optimal composition holds for $\mathrm{COMP}(\mathcal{M}_i)$. That is, if we can prove an $(\varepsilon', \delta')$ upper bound on the privacy parameter for sequential compositions of $\mathcal{M}_1^b, \ldots, \mathcal{M}_k^b$, then the concurrent composition $\mathrm{COMP}(\mathcal{M}_i)$ also enjoys the same $(\varepsilon', \delta')$-DP.

Vadhan and Wang [2021] also considered the case $\delta > 0$ (i.e., approximate DP). However, for this case, they only showed an upper bound on $\varepsilon', \delta'$ that is inferior to the basic composition in the sequential setting. It was asked as an open question in Vadhan and Wang [2021] whether the optimal composition theorem for approximate DP still holds in the concurrent composition.

Besides pure and approximate DP, there are also other notions of differential privacy that are extensively studied in the literature. A non-exhaustive list includes Rényi DP [Mironov, 2017], concentrated DP [Dwork and Rothblum, 2016, Bun and Steinke, 2016, Bun et al., 2018], Gaussian DP and $f$-DP [Dong et al., 2022] etc. Compared with the standard notion of $(\varepsilon, \delta)$-approximate DP, these variants of DP either allow for a simplified analysis of private algorithms or give sharper bounds of privacy guarantee. In the sequential composition, the composition property of these variants has been well understood. It is also interesting to extend these composition theorems to the concurrent composition, thereby expanding the potential applicability of these DP notions.

## 2 Our Results

In this work, we give an affirmative answer to the open question mentioned above. Moreover, our result confirms that several major differential privacy definitions in the literature enjoy the same composition guarantee in the concurrent composition, just as they do in the sequential composition.

**Approximate differential privacy.** $(\varepsilon, \delta)$-DP is arguably the most widely studied notion of differential privacy and is deemed the "standard" definition of DP. As our first main result, we show an optimal concurrent composition theorem for approximate DP.

**Theorem 1.** *Let $\mathcal{M}_1, \ldots, \mathcal{M}_k$ be $k$ interactive mechanisms that run on the same data set. Suppose that each mechanism $\mathcal{M}_i$ satisfies $(\varepsilon_i, \delta_i)$-DP. Then $\mathrm{COMP}(\mathcal{M}_1 \ldots \mathcal{M}_k)$ is $(\varepsilon', \delta')$-DP, where $\varepsilon', \delta'$ are given by the optimal (sequential) composition theorem [Kairouz et al., 2015, Murtagh and Vadhan, 2018].*

*In particular, when the privacy parameter for each mechanism is the same $(\varepsilon, \delta)$, their concurrent composition satisfies $O(\sqrt{k \log(1/\delta')}\varepsilon, \delta' + k\delta)$-DP for all $\delta' \in (0, 1)$.*

**Rényi differential privacy.** Rényi differential privacy was first defined by Mironov [2017]. We recall its definition.

**Definition 3** (Rényi divergence and differential privacy). Let $P, Q$ be two distributions supported over $\mathcal{X}$. For each $\alpha > 1$, define the Rényi divergence of order $\alpha$ of $P$ from $Q$ as

$$D_\alpha(P \| Q) := \frac{1}{\alpha - 1} \log \left( \mathop{\mathbb{E}}_{x \sim P} \left[ \left( \frac{P(x)}{Q(x)} \right)^{\alpha - 1} \right] \right).$$

Two interactive systems are called $(\alpha, \varepsilon)$-Rényi close, if for every adversary $\mathcal{A}$ and every $b \in \{0, 1\}$, it holds that

$$D_\alpha(\mathbf{IT}(\mathcal{A} : \mathcal{M}^b) \| \mathbf{IT}(\mathcal{A} : \mathcal{M}^{1-b})) \leq \varepsilon.$$

Let $\mathcal{M}$ be a mechanism. $\mathcal{M}$ is called $(\alpha, \varepsilon)$-Rényi differentially private (or $(\alpha, \varepsilon)$-RDP for short), if for every two neighboring data sets $d$ and $d'$, the systems $\mathcal{M}^d$ and $\mathcal{M}^{d'}$ are $(\alpha, \varepsilon)$-Rényi close.

A main advantage of Rényi DP is that it has a natural and simple composition. In the sequential setting, it is known that if two mechanisms $\mathcal{M}_1, \mathcal{M}_2$ are $(\alpha, \varepsilon_1)$ and $(\alpha, \varepsilon_2)$-RDP, respectively, then the composition of $\mathcal{M}_1$ and $\mathcal{M}_2$ is $(\alpha, \varepsilon_1 + \varepsilon_2)$-RDP. Our next theorem generalizes this result to the concurrent composition setting.

**Theorem 2.** *Let $\mathcal{M}_1, \ldots, \mathcal{M}_k$ be $k$ interactive mechanisms that run on the same data set. Suppose that each mechanism $\mathcal{M}_i$ is $(\alpha, \varepsilon_i)$-RDP. Then $\mathrm{COMP}(\mathcal{M}_1 \ldots \mathcal{M}_k)$ is $(\alpha, \sum_{i=1}^k \varepsilon_i)$-RDP.*

One implication of Theorem 2 is that the zero-concentrated differential privacy by Bun and Steinke [2016] and the truncated concentrated differential privacy by Bun et al. [2018] also compose nicely under the concurrent composition. We state the corollary below, and prove it in the full version of the paper.

**Corollary 1.** *Let $\mathcal{M}_1, \ldots, \mathcal{M}_k$ are $k$ interactive mechanisms that run on the same data set. Suppose that each mechanism $\mathcal{M}_i$ is $\eta_i$-zCDP (resp. $(\rho_i, \omega)$-tCDP). Then $\mathrm{COMP}(\mathcal{M}_1 \ldots \mathcal{M}_k)$ is $(\sum_i \eta_i)$-zCDP (resp. $(\sum_i \rho_i, \omega)$-tCDP).*

Theorem 1, 2 and Corollary 1 provide compelling evidence that the adversary gains no advantage by interleaving its queries to independently running mechanisms. Consequently, interactivity can be viewed as a feature that differential privacy grants us for free.

**Concurrent and independent work.** Concurrently to our work, a recent work by Vadhan and Zhang [2022] proves an optimal concurrent composition theorem for $f$-DP [Dong et al., 2022]. By the standard connection, their result implies the optimal concurrent composition theorem for approximate DP. However, our techniques are very different than theirs. Their result is stronger, as it is known that approxiamte-DP can be seen as a special case of $f$-DP [Dong et al., 2022]. However, our proof for approximate DP is more elementary: we do not need to work through $f$-DP as their proof does. Furthermore, our proof comes with several interesting technical ingredients that might be of independent interests. This includes a structural result for interactive mechanisms (Theorem 3), as well as a dual perspective to reason about Rényi divergences (Lemma 4).

## 3 Implications of Our Results

In this section, we discuss implications of our results, and demonstrate how they offer more than the sequential composition theorems.

**Designing new algorithms.** The optimal concurrent composition theorem makes it possible to design new differentially private algorithm that involves running several building blocks concurrently. As one motivating example, consider the Sparse Vector Technique (SVT). The standard SVT (as in Dwork and Roth [2014]) and its variants have been studied extensively in the literature. In particular, it was observed by Lyu et al. [2017], Zhu and Wang [2020] that one can add noise to the threshold only *once*, and then use the noisy threshold to answer $c > 1$ "meaningful" queries (namely, after reporting each meaningful query, the SVT algorithm does NOT refresh the noisy threshold). It was argued in [Lyu et al., 2017, Zhu and Wang, 2020] that this variant of SVT can offer a higher accuracy while consuming the same amount of privacy budget, both theoretically and empirically.

However, this variant of SVT has received relatively less attention in literature. One reason might be that it is unclear what happens if we compose this SVT with other mechanisms. In particular, the

standard SVT refreshes its threshold after answering each "meaningful" query, which allows one to decompose the algorithm into $c$ pieces of smaller SVT algorithms, and then compose with other mechanisms via the sequential computation. In contrast, the variants by Lyu et al. [2017], Zhu and Wang [2020] work by answering each "meaningful query" using the *same* noisy threshold, which do not seem to admit such a decomposition. This makes this variant less appealing: in most applications, people want to use SVT as a supporting subroutine for other algorithms. Therefore, it is crucial to understand the (concurrent) composition behavior of SVT with other mechanisms.

Now, with the new concurrent composition theorem, we can plug this variant of SVT in any algorithm, and argue the privacy guarantee of the whole computation by black-box applying Theorems 1 and 2 (depending on whether we are working with $(\varepsilon, \delta)$-DP or RDP). To illustrate the idea, in the full version of the paper, we apply Theorem 1 to analyze a simple algorithm: private "Guess-and-Check" with the aforementioned variant of SVT as a subroutine. We hope our example can motivate people to design more powerful algorithms by concurrently composing simple building blocks.

**Practical Implication.** Besides the theoretical interests, our theorem has implications for practical deployments of interactive DP mechanisms. For one example, suppose there is a data center that holds the private information of individuals and offers data analysts access to the database (interactively and differentially-privately). Without knowing the concurrent composition theorem, it might be possible that some $k > 1$ analysts can collude by coordinating their (interactive) queries to the database and extracting much more sensitive information. Our result refutes the possibility of such an attack. In particular, suppose each data analyst has only an $(\varepsilon, \delta)$-DP amount of privacy "quota". Then, even if they collude and spend their privacy budget in whatever way, their computation result is still $(O(\sqrt{k \log(1/\delta')}\varepsilon), k\delta + \delta')$-DP with respect to the private database.

# 4 Proof of Main Results

In this section, we show the proof of our results. We start with a very brief proof overview. We prove Theorem 1 by a reduction to the sequential composition of $k$ (approximate) randomized response mechanisms. This generalizes the idea developed by Vadhan and Wang [2021]. To prove Theorem 2, we take a completely different approach, and our technique offers new tools to analyze Rényi DP. Namely, we propose an alternative characterization of Rényi divergence (Lemma 4), which allows for a fine-grained account of the privacy loss in the complex interaction involving multiple mechanisms. The characterization of Rényi divergence might find itself useful in other applications.

**Notation.** Let $P, Q$ be two distributions supported over $X$. For a real $\eta > 0$, we say that $P \geq \eta Q$, if for every $S \subseteq X$, it holds that

$$\Pr_{x \sim P}[x \in S] \geq \eta \Pr_{x \sim Q}[x \in S].$$

Furthermore, we say $P \equiv Q$, if $P$ and $Q$ are identically distributed.

## 4.1 Approximate Differential Privacy

To prove Theorem 1, we follow the approach by Vadhan and Wang [2021], where they showed that one can simulate two $(\varepsilon, 0)$-indistinguishable interactive systems by post-processing a randomized response mechanism. This simulation enables them to reduce the concurrent composition to a sequential composition, and the optimal composition theorem follows. It was asked as an open question in Vadhan and Wang [2021] whether the same simulation can be carried out for approximate DP. We answer this question affirmatively.

**Review of the Vadhan-Wang approach.** It would be instructive to review the proof by Vadhan and Wang [2021] first. Let $\mathcal{M}^0, \mathcal{M}^1$ be the pair of systems by running the private mechanism on a pair of neighboring data sets. The adversary $\mathcal{A}$ interacts with $\mathcal{M}^b$ for some $b \in \{0, 1\}$ and wants to find out the value of $b$. An intuitive yet delicate fact due to Vadhan and Wang [2021] is that, if $\mathcal{M}^0$ and $\mathcal{M}^1$ are $(\varepsilon, 0)$-indistinguishable, then there exist two systems $\mathcal{N}^0, \mathcal{N}^1$ such that, for every adversary $\mathcal{A}$, the distribution of $\mathbf{IT}(\mathcal{A} : \mathcal{M}^b)$ is identical to $\frac{e^\varepsilon}{1+\varepsilon}\mathbf{IT}(\mathcal{A} : \mathcal{N}^b) + \frac{1}{1+e^\varepsilon}\mathbf{IT}(\mathcal{A} : \mathcal{N}^{1-b})$. This enables one to simulate the many-round interaction between $\mathcal{A}$ and $\mathcal{M}^b$ by running a one-round randomized response mechanism.

In more detail, let $\mathrm{RR}_\varepsilon^b$ denote the standard randomized response mechanism, defined as follows. $\mathrm{RR}_\varepsilon^b$ only accepts one query. On the query, $\mathrm{RR}_\varepsilon^b$ ignores the query message, returns $b$ with probability $\frac{e^\varepsilon}{1+e^\varepsilon}$, and returns $1-b$ otherwise. We modify $\mathcal{A}$ to a new adversary $\mathcal{A}'$: $\mathcal{A}'$ first sends a query to $\mathrm{RR}_\varepsilon^b$ and receives a bit $b'$. Then $\mathcal{A}'$ simulates the interaction between $\mathcal{A}$ and $\mathcal{N}^{b'}$, and outputs the transcript (i.e., $\mathbf{IT}(\mathcal{A}:\mathcal{N}^{b'})$). Let $\mathbf{Output}(\mathcal{A}':\mathrm{RR}_\varepsilon^b)$ denote the output distribution of $\mathcal{A}'$ when interacting with $\mathrm{RR}_\varepsilon^b$. It is clear that

$$\mathbf{Output}(\mathcal{A}':\mathrm{RR}_\varepsilon^b) \equiv \frac{e^\varepsilon}{1+\varepsilon}\mathbf{IT}(\mathcal{A}:\mathcal{N}^b) + \frac{1}{1+e^\varepsilon}\mathbf{IT}(\mathcal{A}:\mathcal{N}^{1-b}) \equiv \mathbf{IT}(\mathcal{A}:\mathcal{M}^b).$$

Therefore, $\mathcal{A}'$ simulates the interaction between $\mathcal{A}, \mathcal{M}^b$ faithfully, by a single query to $\mathrm{RR}_\varepsilon^b$.

Now, suppose the adversary $\mathcal{A}$ is interacting with $k$ mechanisms $\mathcal{M}_1^b,\ldots,\mathcal{M}_k^b$ in parallel. For each $i \in [k]$, assuming that $\mathcal{M}_i^0$ and $\mathcal{M}_i^1$ are $(\varepsilon_i,0)$-indistinguishable, there is a decomposition of $\mathcal{M}_i^0, \mathcal{M}_i^1$ by some $\mathcal{N}_i^0$ and $\mathcal{N}_i^1$. Again, we modify $\mathcal{A}$ to a new mechanism $\mathcal{A}'$. $\mathcal{A}'$ first queries $\mathrm{RR}_{\varepsilon_i}^b, i \in [k]$ in order, and receives $k$ bits $b_1',\ldots,b_k'$. Then $\mathcal{A}'$ simulates the interaction between $\mathcal{A}$ and $(\mathcal{N}_i^{b_i'})_{i\in[k]}$ and outputs the transcript. One can show that

$$\mathbf{Output}(\mathcal{A}':\mathrm{RR}_{\varepsilon_1}^b,\ldots,\mathrm{RR}_{\varepsilon_k}^b) \equiv \mathbf{IT}(\mathcal{A}:\mathcal{M}_1^b,\ldots,\mathcal{M}_k^b). \tag{2}$$

Note that the left hand side of (2) can be simulated by a sequential composition of $k$ randomized response mechanisms. Invoking the optimal composition theorem for sequential composition [Kairouz et al., 2015, Murtagh and Vadhan, 2018] concludes the proof.

**Extension to approximate DP.** Now, if $\mathcal{M}^0$ and $\mathcal{M}^1$ are $(\varepsilon,\delta)$-indistinguishable with $\delta > 0$, there might not be a nice decomposition of $\mathcal{M}^b$ into $\frac{e^\varepsilon}{1+e^\varepsilon}\mathcal{N}^b + \frac{1}{1+e^\varepsilon}\mathcal{N}^{1-b}$. Still, it is plausible to conjecture that there is a decomposition of $\mathcal{M}^0, \mathcal{M}^1$ with four systems $\mathcal{N}^0, \mathcal{N}^1, \mathcal{E}^0, \mathcal{E}^1$ such that for each $b \in \{0,1\}$,

$$\mathcal{M}^b = \delta\mathcal{E}^b + (1-\delta)\left(\frac{e^\varepsilon}{1+e^\varepsilon}\mathcal{N}^b + \frac{1}{1+e^\varepsilon}\mathcal{N}^{1-b}\right). \tag{3}$$

Our main technical result in this subsection proves the existence of such a decomposition.

**Theorem 3.** *Two systems $\mathcal{M}^0, \mathcal{M}^1$ are $(\varepsilon,\delta)$-indistinguishable, if and only if there are four systems $\mathcal{N}^0, \mathcal{N}^1, \mathcal{E}^0, \mathcal{E}^1$ satisfying the following: for every adversary $\mathcal{A}$ and $b \in \{0,1\}$, it holds that*

$$\mathbf{IT}(\mathcal{A}:\mathcal{M}^b) \equiv \delta\mathbf{IT}(\mathcal{A}:\mathcal{E}^b) + (1-\delta)\left(\frac{e^\varepsilon}{1+\varepsilon}\mathbf{IT}(\mathcal{A}:\mathcal{N}^b) + \frac{1}{1+e^\varepsilon}\mathbf{IT}(\mathcal{A}:\mathcal{N}^{1-b})\right). \tag{4}$$

Theorem 3 implies Theorem 1 by a similar reduction to (approximate) random response. For the full proof, see the full version of the paper.

We prove Theorem 3 by establishing a series of lemmas. In the following, we state these lemmas and explain their intuition. We defer the formal proof to the full version of the paper.

**Lemma 1.** *Suppose $\mathcal{M}^0, \mathcal{M}^1$ are $(\varepsilon,\delta)$-indistinguishable. There are two systems $\mathcal{E}^0, \mathcal{E}^1$ satisfying the following.*

- *For every adversary $\mathcal{A}$ and $b \in \{0,1\}$, it holds that $\mathbf{IT}(\mathcal{A}:\mathcal{M}^b) \geq \delta \cdot \mathbf{IT}(\mathcal{A}:\mathcal{E}^b)$.*

- *For every adversary $\mathcal{A}$, every set of transcripts $S \subseteq \{(x_i,y_i)_{i\in[T]}\}$ and $b \in \{0,1\}$, it holds that*

$$\Pr[\mathbf{IT}(\mathcal{A}:\mathcal{M}^b) \in S] - \delta\Pr[\mathbf{IT}(\mathcal{A}:\mathcal{E}^b) \in S]$$
$$\leq e^\varepsilon\left(\Pr[\mathbf{IT}(\mathcal{A}:\mathcal{M}^{1-b}) \in S] - \delta\Pr[\mathbf{IT}(\mathcal{A}:\mathcal{E}^{1-b}) \in S]\right).$$

Roughly, Lemma 1 says that there are two systems $\mathcal{E}^0, \mathcal{E}^1$ that capture the low-probability "bad behavior" of $\mathcal{M}^0, \mathcal{M}^1$. It is the primary technical contribution of this subsection. We prove Lemma 1 by explicitly constructing the two systems $\mathcal{E}^0, \mathcal{E}^1$. That is, we specify the probability density functions $\Pr[\mathcal{E}^b((x_j,y_j)_{j<i},x_i) = y_i]$ for $\mathcal{E}^0, \mathcal{E}^1$ step by step, in the increasing order of $i = 1,2,\ldots,T$.

**Lemma 2.** *Suppose $\mathcal{M}, \mathcal{E}$ are two systems such that for every adversary $\mathcal{A}$, it holds that $\mathbf{IT}(\mathcal{A} : \mathcal{M}) \geq \delta \mathbf{IT}(\mathcal{A} : \mathcal{E})$. Then there is a system $\mathcal{N}$ such that for every adversary $\mathcal{A}$, it holds that*

$$\mathbf{IT}(\mathcal{A} : \mathcal{M}) \equiv \delta \mathbf{IT}(\mathcal{A} : \mathcal{E}) + (1 - \delta)\mathbf{IT}(\mathcal{A} : \mathcal{N}).$$

For intuition, suppose $P, Q$ are two distributions such that $P \geq \delta Q$. Then one can easily find a distribution $Q'$ such that $P \equiv \delta Q + (1 - \delta)Q'$. The proof of Lemma 2 extends this simple idea.

**Lemma 3** (Vadhan and Wang [2021]). *Suppose $\mathcal{N}^0, \mathcal{N}^1$ are $(\varepsilon, 0)$-indistinguishable. Then there are two systems $\mathcal{N}^{0'}, \mathcal{N}^{1'}$ such that for every adversary $\mathcal{A}$, it holds that*

$$\mathbf{IT}(\mathcal{A} : \mathcal{N}^b) \equiv \frac{e^\varepsilon}{1 + e^\varepsilon}\mathbf{IT}(\mathcal{A} : \mathcal{N}^{b'}) + \frac{1}{1 + e^\varepsilon}\mathbf{IT}(\mathcal{A} : \mathcal{N}^{(1-b)'}).$$

**Wrap-up.** We can conclude the proof for Theorem 3 now. The "if" direction is obvious: the existence of a decomposition satisfying (4) implies that $\mathcal{M}^0, \mathcal{M}^1$ are $(\varepsilon, \delta)$-indistinguishable. For the other direction, we start by constructing $\mathcal{E}^0, \mathcal{E}^1$ using Lemma 1. Then we construct $\mathcal{N}^0, \mathcal{N}^1$ by Lemma 2. Lemma 1 and 2 together ensure that $\mathcal{N}^0$ and $\mathcal{N}^1$ are $(\varepsilon, 0)$-indistinguishable, which enables us to invoke Lemma 3 and decompose $\mathcal{N}^0, \mathcal{N}^1$ into $\mathcal{N}^{0'}, \mathcal{N}^{1'}$. $(\mathcal{E}^0, \mathcal{E}^1, \mathcal{N}^{0'}, \mathcal{N}^{1'})$ forms the final decomposition. It is straightforward to verify that they satisfy (4).

### 4.2 Rényi Differential Privacy

Our result for Rényi differential privacy (Theorem 2) takes a completely different approach.

**An intuition.** Let $\mathcal{M}_1$ be an $(\alpha, \varepsilon)$-Rényi DP mechanism. Intuitively, $(\alpha, \varepsilon)$-Rényi DP means that $\mathcal{M}_1$ has $\varepsilon$ unit of privacy budget and can distribute it to $T$ queries. Viewing the privacy budget as a form of "deposit", we hope to argue that two or more independently running mechanisms spend their deposit independently, and an adversary cannot trigger any mechanism to spend more privacy budget than it holds by interacting with other mechanisms.

However, unlike some intuitive and easy-to-measure resources such as time and energy, the notion of privacy loss looks somewhat illusive. Even worse, we need to reason about this elusive resource in a stochastic process consisting of interactions with multiple systems. It was not clear how one can quantify the privacy loss in such an interactive and complex process. Nonetheless, we manage to find a new approach to do so.

**An alternative characterization for Rényi DP.** We introduce the following characterization of Rényi divergence based on Hölder's inequality and duality. That is, we prove

**Lemma 4** (An alternative characterization of Rényi divergence). *Suppose $P, Q$ are two distributions supported over $\mathcal{Y}$. For every $\alpha > 1$ and $B \geq 0$, let $\beta = \frac{\alpha}{\alpha-1}$ be the Hölder conjugate of $\alpha$. The following statements are equivalent.*

- $D_\alpha(P\|Q) \leq B$.

- *For every function $h : \mathcal{Y} \to \mathbb{R}^{\geq 0}$, it holds that $\mathbb{E}_{y \sim P}[h(y)] \leq e^{\frac{B(\alpha-1)}{\alpha}} \mathbb{E}_{y \sim Q}[h(y)^\beta]^{1/\beta}$.*

Note that if we let $\alpha \to \infty$, then Lemma 4 converges to a characterization of pure-DP. That is, $D_\infty(P\|Q) \leq B$ if and only if $\Pr[P = y] \leq e^B \Pr[Q = y]$ for every $y \in \mathcal{Y}$.

Lemma 4 provides a convenient tool to reason about the privacy loss in an interactive environment consisting of multiple rounds. Intuitively, this is because Condition 2 in the statement above is more amenable to a "hybrid argument". However, to quantify the privacy loss during an interaction, we still need to find a way to track the privacy loss.

**Measure theory setup.** Before we continue, it would be more convenient to switch to a measure-theoretic language. Consider two measures $P, Q$ on a space $\mathcal{Y}$ ($P$ and $Q$ are not necessarily probability measures), we say that $P$ is $\beta$-dominated by $Q$, denoted by $P \preceq_\beta Q$, if for every measurable function $f : \mathcal{Y} \to \mathbb{R}^{\geq 0}$, it holds that

$$\|f\|_{P,1} := \int f(y)dP(y) \leq \left(\int f(y)^\beta dQ(y)\right)^{1/\beta} =: \|f\|_{Q,\beta}.$$

When $\mathcal{Y}$ is a finite set, the integral coincides with an equivalent summation. i.e.,

$$\int f(y)dP(y) = \sum_y P(y)f(y).$$

We will use integral and summation interchangeably.

In this notation, Lemma 4 can be equivalently stated as $D_\alpha(P\|Q) \le B$ if and only if $P$ is $\beta$-dominated by $e^B Q$ for $\beta = \frac{\alpha}{\alpha-1}$ .

The following lemma is essential for us.

**Lemma 5.** *Let $\mathcal{Y}_1 \times \mathcal{Y}_2$ be a space. Consider two distributions $P, Q$ on $\mathcal{Y}_1 \times \mathcal{Y}_2$. Assume $\operatorname{supp}(P) = \operatorname{supp}(Q) = \mathcal{Y}_1 \times \mathcal{Y}_2$. Let $P_1, P_2$ be the margin of $P$ on $\mathcal{Y}_1, \mathcal{Y}_2$. For each $y_1 \in \mathcal{Y}_1$, denote by $P_2|_{P_1=y_1}$ the marginal distribution of $y_2$ conditioning on $y_1$. Also define the same notation for $Q$.*

*Let $\beta \ge 1, B \ge 0$ be two reals. Let $\alpha = \frac{\beta}{\beta-1}$. For each $y_1 \in \mathcal{Y}_1$, define*

$$\ell_1(y_1) = \inf_K \left\{ K : P_2|_{P_1=y_1} \preceq_\beta K \cdot Q_2|_{Q_1=y_1} \right\} = \exp(D_\alpha(P_2|_{P_1=y_1} \| Q_2|_{Q_1=y_1})).$$

*Suppose $P \preceq_\beta e^B Q$. Consider the measure spaces $(\mathcal{Y}_1, P_1(y_1) \cdot \ell_1(y_2)^{1/\beta})$ and $(\mathcal{Y}_1, Q_1)$. We have*

$$P_1 \ell_1^{1/\beta} \preceq_\beta e^B Q_1.$$

Intuitively, the function $\ell(y_1)$ serves as the role of "privacy budget monitor". To see this, fix an adversary $\mathcal{A}$ and think of $(y_1, y_2)$ as the responses of the system to the adversary[2]. After observing $y_1$, the adversary wants to distinguish between two conditional distributions $P_2|_{P_1=y_1}$ and $Q_2|_{Q_1=y_1}$. At this moment, $\ell(y_1)$ shows up as an upper bound of "extra information" that the adversary can extract by utilizing their second query. Alternatively, $\ell(y_1)$ quantifies the amount of the remaining privacy budget the mechanism has after outputting $y_1$. On average, the function $\ell_1(y_1)$ provides a fine-grained control of the privacy loss in the sense that $P_1 \ell_1^{1/\beta} \preceq_\beta e^B Q_1$.

**Proof for a $3$-round toy example.** We are ready to describe the proof for Theorem 2. To illustrate the idea, we prove a toy case here and defer the full proof to the full version of the paper. The proof for the toy case includes all the important ideas. Extending it to a full proof is straightforward.

We describe the toy scenario now. Suppose there are two mechanisms $\mathcal{M}_1, \mathcal{M}_2$ that run on a sensitive input bit $b \in \{0, 1\}$. The interaction consists of 3 rounds. The adversary $\mathcal{A}$ communicates with $\mathcal{M}_1, \mathcal{M}_2, \mathcal{M}_1$ in order, and outputs the response $(y_1, y_2, y_3)$. For brevity, we assume that each response $y_i$ contains a copy of the query message $x_i$, so that we recover the whole transcript $((x_1, y_1), (x_2, y_2), (x_3, y_3))$ only from the responses.

Let $P, Q \in \Delta(\mathcal{Y} \times \mathcal{Y} \times \mathcal{Y})$ be the output distribution when $\mathcal{A}$ interacts with $(\mathcal{M}_1^0, \mathcal{M}_2^0)$ and $(\mathcal{M}_1^1, \mathcal{M}_2^1)$, respectively. Suppose $\mathcal{M}_1, \mathcal{M}_2$ are $(\alpha, \varepsilon_1), (\alpha, \varepsilon_2)$-Rényi DP respectively. Our goal is to prove that

$$\max \left\{ D_\alpha(P\|Q), D_\alpha(Q\|P) \right\} \le \varepsilon_1 + \varepsilon_2.$$

We bound $D_\alpha(P\|Q)$ below. The bound for $D_\alpha(Q\|P)$ is symmetric. Be Lemma 4, it suffices to show that for every $h : \mathcal{Y} \times \mathcal{Y} \to \mathcal{Y} \to \mathbb{R}^{\ge 0}$ that

$$\sum_{y=(y_1,y_2,y_3)} P(y)h(y) \le \left( e^{\varepsilon_1+\varepsilon_2} \sum_{y=(y_1,y_2,y_3)} Q(y)h(y)^\beta \right)^{1/\beta} \tag{5}$$

where $\beta = \frac{\alpha}{\alpha-1}$ is the Hölder conjugate of $\alpha$. Let $P_1, P_2, P_3$ be the projection of $P$ onto the three rounds, and let $P_i|_{y_{<i}}$ denote the distribution of $y_i$ conditioning on $y_1, \ldots, y_{i-1}$. Also define the same notation for $Q$. Then we write

$$\sum_{y=(y_1,y_2,y_3)} P(y)h(y) = \sum_{y_1} \left( P_1(y_1) \sum_{y_2} \left( P_2|_{y_1}(y_2) \sum_{y_3} P_3|_{y_{<3}}(y_3)h(y) \right) \right). \tag{6}$$

---

[2]Although the query made by $\mathcal{A}$ is not explicitly recorded, the pair $(y_1, y_2)$ can capture this information by requiring that each response $y_i$ must be attached with the query message $x_i$. This does not leak any additional information because $x_i$ is solely chosen by $\mathcal{A}$.

For every $y_1 \in \mathcal{Y}$, let $\mathcal{M}_1^0|_{y_1}$ (resp. $\mathcal{M}_1^1|_{y_1}$) denote the interactive system $\mathcal{M}_1^0$ (resp. $\mathcal{M}_1^1$) *conditioning on* that it has answered $y_1$ to the first query (recall we have assumed that $y_1$ contains $x_1$). Formally, for every $b \in \{0, 1\}$, $(x_2, y_2), \ldots, (x_t, y_t)$ and $x_{t+1}$, define

$$\mathcal{M}_1^b|_{y_1}((x_j, y_j)_{2 \leq j \leq t}, x_{t+1}) := \mathcal{M}_1^b((x_j, y_j)_{1 \leq j \leq t}, x_{t+1}).$$

Next, define

$$\ell_1(y_1) := \exp\left( \sup_{A:\text{adversary}} \left\{ D_\alpha\big(\mathbf{IT}(A : \mathcal{M}_1^0|_{y_1}) \| \mathbf{IT}(A : \mathcal{M}_1^1|_{y_1}))\right\}\right). \tag{7}$$

From Lemma 5, one can show that $P_1 \ell_1^{1/\beta} \preceq e^B Q_1$. Turning back to (5), we then have

$$\sum_{y_1}\left( P_1(y_1) \sum_{y_2}\left( P_2|_{y_1}(y_2) \sum_{y_3} P_3|_{y_{<3}}(y_3)\underline{h(y)}\right)\right) \tag{8}$$

$$\leq \sum_{y_1}\left( P_1(y_1) \sum_{y_2}\left( P_2|_{y_1}(y_2) \underline{\left(\ell_1(y_1) \sum_{y_3} Q_3|_{y_{<3}}(y_3)h(y)^\beta\right)^{1/\beta}}\right)\right) \tag{9}$$

$$\leq \sum_{y_1}\left( P_1(y_1) \left(e^{\varepsilon_2} \sum_{y_2}\left( Q_2|_{y_1}(y_2)\ell_1(y_1) \sum_{y_3} Q_3|_{y_{<3}}(y_3)h(y)^\beta\right)\right)^{1/\beta}\right) \tag{10}$$

$$= \sum_{y_1}\left( P_1(y_1)\ell_1(y_1)^{1/\beta} \underline{\left(e^{\varepsilon_2} \sum_{y_2}\left( Q_2|_{y_1}(y_2) \sum_{y_3} Q_3|_{y_{<3}}(y_3)h(y)^\beta\right)\right)^{1/\beta}}\right) \tag{11}$$

$$\leq \left(e^{\varepsilon_1+\varepsilon_2} \sum_{y_1}\left( Q_1(y_1) \sum_{y_2}\left( Q_2|_{y_1}(y_2) \sum_{y_3} Q_3|_{y_{<3}}(y_3)h(y)^\beta\right)\right)\right)^{1/\beta} \tag{12}$$

$$= \left(e^{\varepsilon_1+\varepsilon_2} \sum_y Q(y)h(y)^\beta\right)^{1/\beta}. \tag{13}$$

Here, we used inequalities of the form $\sum_y P(y) \cdot h(y) \leq \left(C \cdot \sum_y Q(y) \cdot h(y)^\beta\right)^{1/\beta}$ three times (they are (8) $\Rightarrow$ (9) $\Rightarrow$ (10) and (11) $\Rightarrow$ (12)). We use underlines to highlight the "$h$" part of each step in the deductions above.

(8) $\Rightarrow$ (9) is the most critical step. To see this, observe that knowing $y_2$ does not change the view of the first mechanism, because the second query is sent to the independently running mechanism $\mathcal{M}_2^b$. Therefore, $\mathcal{M}_1^b|_{y_1}$ remains the same after conditioning on *both* $y_1$ and $y_2$. Now, note that $P_3|_{y_{<3}}$ (resp. $Q_3|_{y_{<3}}$) exactly describes one round of interaction between the adversary and $\mathcal{M}_1^0|_{y_1}$ (resp. $\mathcal{M}_1^1|_{y_1}$). Consequently, the information leaked by $y_3$ must be subject to the bound (7) and the inequality holds. Having verified (8) $\Rightarrow$ (9), the steps (9) $\Rightarrow$ (10) and (11) $\Rightarrow$ (12) are straightforward.

Having justified (13) for every measure function $h$, we conclude that $D_\alpha(P\|Q) \leq e^{\varepsilon_1+\varepsilon_2}$. A symmetric argument shows that $D_\alpha(Q\|P) \leq e^{\varepsilon_1+\varepsilon_2}$. This completes the proof for the toy example.

**Proof sketch for the general case.** The proof for the general case extends the idea above with some minor twists. By induction, we only need to prove the composition theorem for the case with two mechanisms and many rounds. An issue worth noting is that $\mathcal{A}$ can choose the next query object based on previous responses. However, we can suppose without loss of generality that $\mathcal{A}$ always communicates with mechanisms alternately, by adding a vanilla query $x^*$ to the query space. If the current mechanism is not the one $\mathcal{A}$ wishes to speak with, $\mathcal{A}$ just sends the vanilla query $x^*$. The mechanism then returns a fixed response, which does not leak any information.

## 5   Conclusion and Future Directions

In this work, we consider the concurrent composition of interactive mechanisms. Regarding the general privacy guarantee under the concurrent composition, our result gives optimal composition

theorems for several popular definitions of differential privacy, including $(\varepsilon, \delta)$-DP and Rényi DP. Our work is purely theoretical, and we do not see any negative societal impacts it may cause.

For future directions, we ask whether one can use our composition theorems to design new differentially-private algorithms that may involve running several differentially-private mechanisms in parallel. It is also interesting to explore more practical implications of the concurrent composition phenomena.

We also note that there is a recent interest in *fully adaptive* compositions of differential privacy, which studies how the data analyst can manage the privacy budget and monitor the privacy loss themselves. In particular, the notion of privacy odometers and filters were proposed to capture these demands Rogers et al. [2016], Feldman and Zrnic [2021], Whitehouse et al. [2022], Lécuyer [2021]. This question necessitates a better understanding of information leakage in an interactive environment. Our work developed several new tools and techniques to reason about interactive mechanisms. Can our technique be useful in studying fully adaptive compositions?

## Acknowledgements

I am grateful to Jelani Nelson for advising this project and providing useful comments on an early draft of this paper. I would also like to thank Salil Vadhan and Wanrong Zhang for insightful discussions about their work.

X. Lyu was supported by ONR DORECG award N00014-17-1-2127.

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
