# A    Appendix: Missing Proofs

In this appendix, we show the formal proofs for all the lemmas and claims in the main paper.

## A.1    Proofs for Approximate Differential Privacy

This subsection present omitted proofs in Section 4.1.

### A.1.1    The key lemma

We start with the proof for Lemma 1.

**Reminder of Lemma 1.** *Suppose $\mathcal{M}^0, \mathcal{M}^1$ are $(\varepsilon, \delta)$-indistinguishable. There are two systems $\mathcal{E}^0, \mathcal{E}^1$ satisfying the following.*

- *For every adversary $\mathcal{A}$ and $b \in \{0,1\}$, it holds that $\mathbf{IT}(\mathcal{A} : \mathcal{M}^b) \geq \delta \cdot \mathbf{IT}(\mathcal{A} : \mathcal{E}^b)$.*

- *For every adversary $\mathcal{A}$, every set of transcripts $S \subseteq \{(x_i, y_i)_{i \in [T]}\}$ and $b \in \{0,1\}$, it holds that*

$$\Pr[\mathbf{IT}(\mathcal{A} : \mathcal{M}^b) \in S] - \delta \Pr[\mathbf{IT}(\mathcal{A} : \mathcal{E}^b) \in S]$$
$$\leq e^{\varepsilon} \left( \Pr[\mathbf{IT}(\mathcal{A} : \mathcal{M}^{1-b}) \in S] - \delta \Pr[\mathbf{IT}(\mathcal{A} : \mathcal{E}^{1-b}) \in S] \right).$$

*Proof.* Without loss of generality, we assume that the interaction between $\mathcal{M}^{0/1}$ and $\mathcal{A}$ consists of exactly $T \in \mathbb{N}$ rounds. For every $t \in [T]$, each $(x_i, y_i)_{i \in [t]}$ and $b \in \{0,1\}$, denote

$$M^b((y_i)_{i \in [t]}, (x_i)_{i \in [t]}) := \prod_{i=1}^{t} \Pr[\mathcal{M}^b((x_j, y_j)_{j<i}, x_i) = y_i]. \tag{14}$$

Intuitively, $M^b((y_i)_{i \in [t]}, (x_i)_{i \in [t]})$ is the probability of $\mathcal{M}^b$ responding $(y_1, \ldots, y_t)$, conditioning on that the query messages are fixed to $(x_1, \ldots, x_t)$. Note that knowing $M^b((y_i)_{i \in [T]}, (x_i)_{i \in [T]})$ for every $(x_i, y_i)_{i \in [T]}$ *uniquely* determines the system.

Let $\mathcal{A}$ be an arbitrary adversary. For each $(x_i, y_i)_{i \in [t-1]}$ and $x_t$, denote

$$A((x_i)_{i \in [t]}, (y_i)_{i \in [t-1]}) := \prod_{i=1}^{t} \Pr[\mathcal{A}((x_j, y_j)_{j<i}) = x_i]. \tag{15}$$

Note that $A((x_i)_{i \in [t]}, (y_i)_{i \in [t-1]})$ is the probability of $\mathcal{A}$ sending queries $(x_1, \ldots, x_t)$, conditioning on that the responses to the first $t-1$ queries are fixed to $(y_1, \ldots, y_{t-1})$.

In the following, when the size of a list $(\ell_i)_{i \in [L]}$ is clear from context, we may omit the subscript and simply write $(\ell_i)$ to denote the list. Now, having defined (14) and (15), we observe for each transcript $(x_i, y_i)_{i \in [T]}$ that

$$\Pr[\mathbf{IT}(\mathcal{A}, \mathcal{M}^b) = (x_i, y_i)_{i \in [T]}] = M^b((y_i), (x_i)) \cdot A((x_i), (y_i)). \tag{16}$$

In the following, we will construct two systems $\mathcal{E}^{0/1}$ such that, for every $(x_i, y_i)_{i \in [T]}$, it holds that

$$M^b((y_i), (x_i)) \geq \delta E^b((y_i), (x_i)) \tag{17}$$

and

$$M^b((y_i),(x_i)) - \delta E^b((y_i),(x_i)) \le e^\varepsilon \left( M^{1-b}((y_i),(x_i)) - \delta E^{1-b}((y_i),(x_i)) \right). \tag{18}$$

If we have two systems $\mathcal{E}^{0/1}$ satisfying the above, then we can verify that they satisfy the lemma statement by combining (16), (17) and (18).

Now we describe the construction. We start by defining for each $b \in \{0,1\}$, $t \le T$ and every partial history $(x_i, y_i)_{i \le t-1} \in (\mathcal{X} \times \mathcal{Y})^{t-1}$, $x_t \in \mathcal{X}$ a control function as

$$\mathsf{Lower}^b((x_i, y_i)_{i<t}, x_t) := \begin{cases} \sum_{y_t \in \mathcal{Y}} \max_{x_{t+1}} \{\mathsf{Lower}^b((x_i, y_i)_{i \le t}, x_{t+1})\} & t < T \\ \sum_{y_t \in \mathcal{Y}} \max\left\{ M^b((y_i),(x_i)) - e^\varepsilon M^{1-b}((y_i),(x_i)), 0 \right\} & t = T \end{cases}. \tag{19}$$

For every $t \le T-1$ and $(x_i, y_i)_{i \le t}$, we also define the following control function:

$$\mathsf{Upper}^b((x_i, y_i)_{i \le t}) := M^b((y_i)_{i \le t},(x_i)_{i \le t}) - e^{-\varepsilon} M^{1-b}((y_i)_{i \le t},(x_i)_{i \le t}). \tag{20}$$

We need the following two facts regarding the control functions.

**Claim 1.** *For each $b \in \{0,1\}$ and $x_1 \in \mathcal{X}$, it holds that $\mathsf{Lower}^b(\emptyset, x_1) \le \delta$.*

*Proof.* We construct an adversary $\mathcal{A}^*$ as follows. $\mathcal{A}^*$ is deterministic. It always sends $x_1$ as the first query. For every $1 \le t \le T-1$ and history $(x_i, y_i)_{i \in [t]}$, $\mathcal{A}^*$ computes the next query as

$$\mathcal{A}^*((x_i, y_i)_{i \in [t]}) = \arg\max_{x_{t+1}} \{\mathsf{Lower}^b((x_i, y_i)_{i \le t}, x_{t+1})\}.$$

Now, define

$$S^b := \{(x_i, y_i)_{i \in [T]} : \Pr[\mathbf{IT}(\mathcal{A}^*, \mathcal{M}^b) = (x_i, y_i)_{i \in [T]}] > e^\varepsilon \Pr[\mathbf{IT}(\mathcal{A}^*, \mathcal{M}^{1-b}) = (x_i, y_i)_{i \in [T]}]\}.$$

Given that $\mathcal{M}^0$ and $\mathcal{M}^1$ are $(\varepsilon, \delta)$-indistinguishable, we know that

$$\sum_{(x_i, y_i) \in S^b} \Pr[\mathbf{IT}(\mathcal{A}^*, \mathcal{M}^b) = (x_i, y_i)_{i \in [T]}] - e^\varepsilon \Pr[\mathbf{IT}(\mathcal{A}^*, \mathcal{M}^{1-b}) = (x_i, y_i)_{i \in [T]}] \le \delta.$$

On the other hand, by the definition of $\mathcal{A}^*$ and (19), it holds that

$$\mathsf{Lower}^b(\emptyset, x_1) = \sum_{(x_i, y_i) \in S^b} \Pr[\mathbf{IT}(\mathcal{A}^*, \mathcal{M}^b) = (x_i, y_i)] - e^\varepsilon \Pr[\mathbf{IT}(\mathcal{A}^*, \mathcal{M}^{1-b}) = (x_i, y_i)].$$

This can be verified by tracing how $\mathsf{Lower}^b(\emptyset, x_1)$ is determined from queries $(x_1, \ldots, x_T)$ (in the "max" operator), and noting that $\mathcal{A}^*$ follows exactly the same queries. Combining two equations above concludes the proof of Claim. $\square$

**Claim 2.** *For every $b \in \{0,1\}$, $t \le T-1$ and $(x_i, y_i)_{i \le t-1} \in (\mathcal{X} \times \mathcal{Y})^{t-1}$, $x_t \in \mathcal{X}$, it holds that*

$$\mathsf{Lower}^b((x_i, y_i)_{i<t}, x_t) \le \mathsf{Upper}^b((x_i, y_i)_{i<t}) + e^{-\varepsilon} \mathsf{Lower}^{1-b}((x_i, y_i)_{i<t}, x_t).$$

*Proof.* We prove this claim by *downward* induction on $t$. For the case $t = T-1$, we have by definition that

$$e^{-\varepsilon} \mathsf{Lower}^{1-b}((x_i, y_i)_{i<t}, x_t)$$

$$= \sum_{y_t \in \mathcal{Y}} \max\left\{ e^{-\varepsilon} M^{1-b}((y_i),(x_i)) - M^b((y_i),(x_i)), 0 \right\} \tag{21}$$

$$= \sum_{y_t \in \mathcal{Y}} e^{-\varepsilon} M^{1-b}((y_i),(x_i)) - M^b((y_i),(x_i)) +$$

$$\sum_{y_t \in \mathcal{Y}} \max\left\{ M^b((y_i),(x_i)) - e^{-\varepsilon} M^{1-b}((y_i),(x_i)), 0 \right\} \tag{22}$$

$$\ge -\mathsf{Upper}^b((x_i, y_i)_{i<t}) + \mathsf{Lower}^b((x_i, y_i)_{i<t}, x_t). \tag{23}$$

We justify the deductions briefly. (21) is by definition. (22) uses a simple trick that $\max\{a, 0\} = a + \max\{-a, 0\}$. The last step (23) is by definition again. In particular, we observe that for every $x_T \in \mathcal{X}$, it holds that

$$\sum_{y_T \in \mathcal{Y}} M^b((y_i), (x_i)) = \prod_{i=1}^{T-1} \Pr[\mathcal{M}^b((x_j, y_j)_{j<i}, x_i) = y_i].$$

This proves the base case for $t = T - 1$.

Assume the claim is true for $t + 1 \leq T - 1$. We consider the case of $t$. We have

$$\mathsf{Lower}^b((x_i, y_i)_{i<t}, x_t) = \sum_{y_t} \max_{x_{t+1}}\{\mathsf{Lower}^b((x_i, y_i)_{i\leq t}, x_{t+1})\}$$

$$\leq \sum_{y_t} \max_{x_{t+1}}\{\mathsf{Upper}^b((x_i, y_i)_{i\leq t}) + e^{-\varepsilon}\mathsf{Lower}^{1-b}((x_i, y_i)_{i\leq t}, x_{t+1})\}$$

$$\leq \sum_{y_t} \mathsf{Upper}^b((x_i, y_i)_{i\leq t}) + e^{-\varepsilon} \max_{x_{t+1}}\{\mathsf{Lower}^{1-b}((x_i, y_i)_{i\leq t}, x_{t+1})\}$$

$$= \mathsf{Upper}^b((x_i, y_i)_{i<t}) + e^{-\varepsilon}\mathsf{Lower}^{1-b}((x_i, y_i)_{i<t}, x_t).$$

The first inequality is due to the induction hypothesis. The second inequality is straightforward. This completes the proof for the claim. □

**The construction.** We are ready to describe the construction. In the following, we will assume $\varepsilon > 0$. Having shown the construction for every $\varepsilon > 0$, the case for $\varepsilon = 0$ can be argued by continuity. We will construct $\mathcal{E}^0, \mathcal{E}^1$ by specifying for every $t \in [T]$ and $(x_i, y_i)_{i\leq t}$ the following:

$$E^b((y_i)_{i\leq t}, (x_i)_{i\leq t}) := \prod_{i=1}^{t} \Pr[\mathcal{E}^b((x_j, y_j)_{j<i}, x_i) = y_i].$$

Note that a valid $E^b(\cdot)$ uniquely defines a system $\mathcal{E}^b$. For brevity, we also define $E^0(\emptyset) = E^1(\emptyset) = 1$. Intuitively, we use $\emptyset$ to denote two "empty lists" (i.e., two lists $(y_i)_{i\leq t}, (x_i)_{i\leq t}$ with $t = 0$).

We will construct $E^b((y_i)_{i\leq t}, (x_i)_{i\leq t})$ for $t = 1, 2, \ldots, T$ in order. Throughput the construction, we maintain the following property. For every $0 \leq t \leq T$, $(x_i, y_i)_{i\leq t}$ and $b \in \{0, 1\}$, we require

$$\delta \cdot E^b((y_i)_{i\leq t}, (x_i)_{i\leq t}) \geq \begin{cases} \max_{x_{t+1}\in\mathcal{X}}\{\mathsf{Lower}^b((x_j, y_j)_{j\leq t}, x_{t+1})\} & t < T \\ \max\{M^b((y_i), (x_i)) - e^{\varepsilon}M^{1-b}((y_i), (x_i)), 0\} & t = T \end{cases} \quad (24)$$

and

$$\delta \cdot E^b((y_i)_{i\leq t}, (x_i)_{i\leq t}) \leq \mathsf{Upper}^b((x_i, y_i)_{i\leq t}) + e^{-\varepsilon}\delta \cdot E^{1-b}((y_i)_{i\leq t}, (x_i)_{i\leq t}). \quad (25)$$

Meanwhile, for $E^b((y_i), (x_i))$ to describe a valid system, it is necessary and sufficient for it to be non-negative and satisfy the following equation for every $(x_i, y_i)_{i\leq t} \in (\mathcal{X} \times \mathcal{Y})^t$ and $x_{t+1}$:

$$\sum_{y_{t+1}\in\mathcal{Y}} E^b((y_i)_{i\leq t+1}, (x_i)_{i\leq t+1}) = E^b((y_i)_{i\leq t}, (x_i)_{i\leq t}). \quad (26)$$

Next, we shall prove that we can construct a valid $E^{0/1}$ satisfying (24), (25) and (26). As we have said, we will construct $E^b$ gradually in the increasing order of $t \in [T]$. For $t = 0$, we have set $E^0(\emptyset) = E^1(\emptyset) = 1$. (24) holds by Claim 1, and (25) holds trivially.

Now let $t < T$. Also let $(y_i)_{i\leq t} \in \mathcal{Y}^t, (x_i)_{i\leq t} \in \mathcal{X}^t$ be two lists. Suppose we have constructed $E^{0/1}((y_i)_{i\leq t}, (x_i)_{i\leq t})$ that satisfies (24) and (25). For every $x_{t+1} \in \mathcal{X}$ and $y_{t+1} \in \mathcal{Y}$, we construct $E^{0/1}((y_i)_{i\leq t+1}, (x_i)_{i\leq t+1})$ in the following.

Fix $x_{t+1} \in \mathcal{X}$. We temporarily set

$$\widetilde{E}^b((y_i)_{i\leq t+1}, (x_i)_{i\leq t+1}) = \frac{1}{\delta} \cdot \begin{cases} \max_{x_{t+2}\in\mathcal{X}}\{\mathsf{Lower}^b((x_j, y_j)_{j\leq t+1}, x_{t+2})\} & t + 1 < T \\ \max\{M^b((y_i), (x_i)) - e^{\varepsilon}M^{1-b}((y_i), (x_i)), 0\} & t + 1 = T \end{cases}.$$

By Claim 2, we know that $\widetilde{E}^b$ satisfies (25). By the construction, $\widetilde{E}^b$ satisfies (24). However, $\widetilde{E}^b$ may fail to satisfy (26). Still, we have

$$\sum_{y_{t+1}} \widetilde{E}^b((y_i)_{i\leq t+1}, (x_i)_{i\leq t+1}) \leq \frac{1}{\delta} \mathsf{Lower}^b((x_i, y_i)_{i\leq t}, x_{t+1}) \leq E^b((y_i)_{i\leq t}, (x_i)_{i\leq t}).$$

In the following, we show that one can adjust $\widetilde{E}^b$ by increasing some $\widetilde{E}^b((y_i)_{i\leq t+1}, (x_i)_{i\leq t+1})$ properly, so that the new $\widetilde{E}^b$ satisfies all of (24), (25) and (26).

To begin with, we define for each $b \in \{0, 1\}$ the quantity

$$\mathrm{Gap}_b := E^b((y_i)_{i\leq t}, (x_i)_{i\leq t}) - \sum_{y_{t+1}} \widetilde{E}^b((y_i)_{i\leq t+1}, (x_i)_{i\leq t+1}). \tag{27}$$

Our goal is to decrease $\mathrm{Gap}_0, \mathrm{Gap}_1$ to zero by increasing $\widetilde{E}$. Since we only increase $\widetilde{E}^b((y_i)_{i\leq t+1}, (x_i)_{i\leq t+1})$, (24) can never be compromised and we only need to consider (25). Consider $y_{t+1}$ and $b \in \{0, 1\}$. We say that $\widetilde{E}^b$ is *tight* at $y_{t+1}$, if

$$\delta\widetilde{E}^b((y_i)_{i\leq t+1}, (x_i)_{i\leq t+1}) = \mathsf{Upper}^b((x_i, y_i)_{i\leq t+1}) + e^{-\varepsilon}\delta\widetilde{E}^{1-b}((y_i)_{i\leq t+1}, (x_i)_{i\leq t+1}).$$

Intuitively, $\widetilde{E}^b$ being tight at $y_{t+1}$ means that we cannot increase $\widetilde{E}^b((y_i)_{i\leq t+1}, (x_i)_{i\leq t+1})$ without increasing $\widetilde{E}^{1-b}((y_i)_{i\leq t+1}, (x_i)_{i\leq t+1})$.

Here shows our adjustment strategy. We consider each $y_{t+1} \in \mathcal{Y}$ in an arbitrary but fixed order. For each $y_{t+1}$, we gradually increase $\widetilde{E}^{0/1}((y_i)_{i\leq t+1}, (x_i)_{i\leq t+1})$ until one of the following events happens.

- Both $\widetilde{E}^0$ and $\widetilde{E}^1$ get tight at $y_{t+1}$.

- $\mathrm{Gap}_0 = 0$, and $\widetilde{E}^1$ is tight at $y_{t+1}$.

- $\mathrm{Gap}_1 = 0$, and $\widetilde{E}^0$ is tight at $y_{t+1}$.

- $\mathrm{Gap}_0 = \mathrm{Gap}_1 = 0$.

It is easy to see that if none of the above happens, we can keep increasing $\widetilde{E}^{0/1}$ at $y_{t+1}$[3]. This completes the description of the adjustment strategy.

Now, we claim that after the adjustment, we must have $\mathrm{Gap}_0 = \mathrm{Gap}_1 = 0$. Suppose it is not the case. For example, suppose $\mathrm{Gap}_0 \neq 0$. Then we know that $\widetilde{E}^0$ is tight at every $y_{t+1}$. This means that

$$\sum_{y_{t+1}} \delta\widetilde{E}^0((y_i)_{i\leq t+1}, (x_i)_{i\leq t+1}) = \mathsf{Upper}^0((x_i, y_i)_{i\leq t}) + e^{-\varepsilon}\sum_{y_{t+1}} \delta\widetilde{E}^1((y_i)_{i\leq t+1}, (x_i)_{i\leq t+1}).$$

Recall that

$$\delta E^0((y_i)_{i\leq t}, (x_i)_{i\leq t}) \leq \mathsf{Upper}^0((x_i, y_i)_{i\leq t}) + e^{-\varepsilon}\delta E^1((y_i)_{i\leq t}, (x_i)_{i\leq t}).$$

Subtracting the inequality with the equality above, we deduce that $\mathrm{Gap}_0 \leq e^{-\varepsilon}\mathrm{Gap}_1$. It implies that $\mathrm{Gap}_1 > 0$. And we can use a symmetric argument to show that $\mathrm{Gap}_1 \leq e^{-\varepsilon}\mathrm{Gap}_0$. Since $e^{-\varepsilon} < 1$, the only solution to the system of inequalities is $\mathrm{Gap}_0 = \mathrm{Gap}_1 = 0$, a contradiction!

Having proven the claim, we know there is a way to adjust $\widetilde{E}^{0/1}$ so that they satisfy all of (24), (25), (26). We then set $E^{0/1}((y_i)_{i\leq t+1}, (x_i)_{i\leq t+1})$ to be $\widetilde{E}^{0/1}((y_i)_{i\leq t+1}, (x_i)_{i\leq t+1})$ and finish the construction for $(x_i, y_i)_{i\leq t}$ and $x_{t+1}$.

We use the construction above for $t = 0, 1, \ldots, T-1$ in order to construct $E^{0/1}$. It remains to verify that $E^{0/1}$ satisfies the lemma statement. It suffices to verify for every $(x_i, y_i)_{i\leq T} \in (\mathcal{X} \times \mathcal{Y})^T$ and $b \in \{0, 1\}$ that

$$M^b((y_i), (x_i)) \geq \delta E^b((y_i), (x_i))$$

---

[3]To see this, note that for $b \in \{0, 1\}$, we can keep increasing $\widetilde{E}^b$ until either (1) $\widetilde{E}^b$ gets tight at $y_{t+1}$, or (2) $\mathrm{Gap}_b = 0$. Therefore, if we cannot increase both $\widetilde{E}^0$ and $\widetilde{E}^1$, it must be one of the four cases above.

and

$$(M^b((y_i),(x_i)) - \delta E^b((y_i),(x_i))) \leq e^{\varepsilon}(M^{1-b}((y_i),(x_i)) - \delta E^{1-b}((y_i),(x_i))).$$

In fact, since $e^{\varepsilon} > 1$, it suffices to verify the second inequality for $b \in \{0,1\}$. This can be verified by utilizing (25): note that $\mathsf{Upper}^b((x_i,y_i)_{i \leq T}) = M^b((y_i),(x_i)) - e^{-\varepsilon}M^{1-b}((y_i),(x_i))$, and (25) tells us that

$$\delta E^b((y_i),(x_i)) \leq M^b((y_i),(x_i)) - e^{-\varepsilon}M^{1-b}((y_i),(x_i)) + e^{-\varepsilon}\delta E^{1-b}((y_i),(x_i)).$$

Re-arranging proves the desired inequality. □

**Remark 1.** *Note that to verify the correctness of $E^{0/1}$, we only used the condition* (25)*. It seems that* (24) *is useless in this proof. However, note that it is possible that* $\mathsf{Upper}^b((y_i),(x_i))$ *is negative for some* $(x_i,y_i)_{i \leq t}$*, which makes it unclear whether* (25) *can always be satisfied by a positive valuation of $E$. This is why we need the other control function* $\mathsf{Lower}$*.*

### A.1.2 Wrap-up

Next, we quickly prove Lemma 2.

**Reminder of Lemma 2.** *Suppose $\mathcal{M}, \mathcal{E}$ are two systems such that for every adversary $\mathcal{A}$, it holds that $\mathbf{IT}(\mathcal{A} : \mathcal{M}) \geq \delta\mathbf{IT}(\mathcal{A} : \mathcal{E})$. Then there is a system $\mathcal{N}$ such that for every adversary $\mathcal{A}$, it holds that*

$$\mathbf{IT}(\mathcal{A} : \mathcal{M}) \equiv \delta\mathbf{IT}(\mathcal{A} : \mathcal{E}) + (1-\delta)\mathbf{IT}(\mathcal{A} : \mathcal{N}).$$

*Proof.* We follow the notation in Section A.1.1. Namely, for each $(x_i,y_i)_{i \leq t}$, define

$$M((y_i)_{i \leq t},(x_i)_{i \leq t}) = \prod_{i=1}^{t} \Pr[\mathcal{M}((x_j,y_j)_{j < i},x_i) = y_i].$$

Also define the same notation for $E$. Then we construct

$$N((y_i)_{i \leq t},(x_i)_{i \leq t}) = \frac{1}{1-\delta}\left(M((y_i)_{i \leq t},(x_i)_{i \leq t}) - \delta E((y_i)_{i \leq t},(x_i)_{i \leq t})\right).$$

Since $\mathcal{M} \geq \delta\mathcal{E}$, we know that $N((y_i),(x_i))$ is always non-negative. Moreover, $N$ encodes a valid system because

$$\sum_{y_{t+1} \in \mathcal{Y}} N((y_i)_{i \leq t+1},(x_i)_{i \leq t+1})$$
$$= \frac{1}{1-\delta}\sum_{y_{t+1} \in \mathcal{Y}} M((y_i)_{i \leq t+1},(x_i)_{i \leq t+1}) - \delta E((y_i)_{i \leq t+1},(x_i)_{i \leq t+1})$$
$$= \frac{1}{1-\delta}\left(M((y_i)_{i \leq t},(x_i)_{i \leq t}) - \delta E((y_i)_{i \leq t},(x_i)_{i \leq t})\right)$$
$$= N((y_i)_{i \leq t},(x_i)_{i \leq t}).$$

Finally, it is easy to verify $\mathbf{IT}(\mathcal{A} : \mathcal{M}) \equiv \delta\mathbf{IT}(\mathcal{A} : \mathcal{E}) + (1-\delta)\mathbf{IT}(\mathcal{A} : \mathcal{M})$. □

As we have shown in Section 4.1, combining Lemma 1, 2 and 3 together, we can prove Theorem 3 easily. Next, we show how Theorem 3 implies Theorem 1.

*Proof of Theorem 1.* Let $\mathcal{M}_1, \ldots, \mathcal{M}_k$ be $k$ mechanisms, where $\mathcal{M}_i$ is $(\varepsilon_i, \delta_i)$-approximate differentially private. We assume without loss of generality that all of $\mathcal{M}_i$'s hold a bit $b \in \{0,1\}$ as the sensitive data.

Let $\mathcal{A}$ be an arbitrary adversary interacting with $\mathrm{COMP}(\mathcal{M}_1, \ldots, \mathcal{M}_k)$. Next, we show how one can simulate $\mathbf{IT}(\mathcal{A}, \mathrm{COMP}(\mathcal{M}_1^b, \ldots, \mathcal{M}_k^b))$ by running $k$ (approximate) randomized response

mechanisms. For each $\mathcal{M}_i^b$, construct an approximate randomized response mechanism $\mathrm{RR}_{\varepsilon_i,\delta_i}^b$. The output distribution of $\mathrm{RR}_{\varepsilon_i,\delta_i}^b$ is:

$$\mathrm{RR}_{\varepsilon_i,\delta_i}^b = \begin{cases} (b, \top) & \text{w.p. } \delta \\ (b, \bot) & \text{w.p. } (1-\delta)\frac{e^\varepsilon}{1+e^\varepsilon} \\ (1-b, \bot) & \text{w.p. } (1-\delta)\frac{1}{1+e^\varepsilon} \end{cases}.$$

We also prepare the decomposition of $\mathcal{M}_i^{0/1}$ with $\mathcal{N}_i^{0/1}, \mathcal{E}_i^{0/1}$ as promised by Theorem 3.

Now, we construct a simulator $\mathcal{S}$ as follows. For each $i \in [k]$, $\mathcal{S}$ runs $\mathrm{RR}_{\varepsilon_i,\delta_i}^b$ and gets a pair $(b_i, \sigma_i)$. If $\sigma_i = \top$, then let $\mathcal{B}_i \leftarrow \mathcal{E}_i^{b_i}$. Otherwise, let $\mathcal{B}_i \leftarrow \mathcal{N}_i^{b_i}$. In this way, $\mathcal{S}$ gets a list of $k$ systems $(\mathcal{B}_1, \ldots, \mathcal{B}_k)$. The simulator then simulates the interaction between $\mathcal{A}$ and $\mathcal{B}_1, \ldots, \mathcal{B}$, and outputs the interaction history. Let $\mathbf{Output}(\mathcal{S}, b)$ denote the output distribution of $\mathcal{S}$. We claim that

$$\mathbf{Output}(\mathcal{S}, b) \equiv \mathbf{IT}(\mathcal{A} : \mathcal{M}_1^b, \ldots, \mathcal{M}_k^b). \tag{28}$$

To see this, for each $\mathcal{M}_i^b$, consider a two-party communication, where one party is $\mathcal{M}_i^b$, and the other party consists of $\mathcal{A}$ and $\mathcal{M}_j^b$ for $j \neq i$. The second party simulates all the interactions between $\mathcal{A}$ and $\mathcal{M}_j^b$, and only sends queries to $\mathcal{M}_i^b$ when $\mathcal{A}$ queries $\mathcal{M}_i^b$. From the second party's viewpoint, $\mathcal{M}_i^b$ looks identical to $\delta_i \mathcal{E}_i^b + (1-\delta_i)\frac{e^\varepsilon}{1+e^\varepsilon}\mathcal{N}_i^b + (1-\delta_i)\frac{1}{1+e^\varepsilon}\mathcal{N}_i^{1-b}$. Therefore,

$$\mathbf{IT}(\mathcal{A} : \mathcal{M}_1^b, \ldots, \mathcal{M}_i^b, \ldots, \mathcal{M}_k^b) \equiv \sum_{j=1}^{3} p_j \cdot \mathbf{IT}(\mathcal{A} : \mathcal{M}_1^b, \ldots, \mathcal{M}_{i,j}^b, \ldots, \mathcal{M}_k^b).$$

Here, we use $(p_1, p_2, p_3) = (\delta_i, (1-\delta_i)\frac{e^\varepsilon}{1+e^\varepsilon}, (1-\delta_i)\frac{1}{1+e^\varepsilon})$ and $(\mathcal{M}_{i,1}^b, \mathcal{M}_{i,2}^b, \mathcal{M}_{i,3}^b) = (\mathcal{E}_i^b, \mathcal{N}_i^b, \mathcal{N}_i^{1-b})$ for convenience. Applying this decomposition for every $i \in [k]$ proves (28).

Finally, note that $\mathbf{Output}(\mathcal{S}, b)$ is just a post-processing of the sequential composition of $k$ (approximate) randomized response mechanisms. Hence, the optimal sequential composition theorem holds for $\mathbf{Output}(\mathcal{S}, b)$, which completes the proof. $\square$

## A.2 Proofs for Rényi Differential Privacy

In this section, we show omitted proofs for Theorem 2.

### A.2.1 Preliminaries

We need some technical preparations first. Consider a measure space $(X, \mu)$. For two measurable functions $f, g$, define their inner product as

$$\langle f, g \rangle_\mu = \int f \cdot g d\mu.$$

For a real $\alpha \geq 1$, define the $\ell_\alpha$-norm of a function $f$ as

$$\|f\|_{\mu,\alpha} := \left( \int f^\alpha d\mu \right)^{1/\alpha}.$$

Recall Hölder's inequality, which is essential for our proof.

**Fact 1.** *Suppose $\alpha, \beta \geq 1$ are Hólder conjugates of each other (i.e., $\frac{1}{\alpha} + \frac{1}{\beta} = 1$). Suppose $f, g$ are two measurable functions. Then we have*

$$\langle f, g \rangle_\mu \leq \|f\|_{\mu,\alpha} \cdot \|g\|_{\mu,\beta}.$$

*The inequality is sharp in the sense that for every measurable function $f$, we have*

$$\|f\|_{\mu,\alpha} = \sup_{h : h \not\equiv 0} \frac{\langle f, h \rangle_\mu}{\|h\|_{\mu,\beta}}.$$

Recall our definition of dominance. For two measures $P, Q$ on a space $\mathcal{Y}$, we say that $P$ is $\beta$-dominated by $Q$, denoted by $P \preceq_\beta Q$, if for every measurable function $f : \mathcal{Y} \to \mathbb{R}^{\geq 0}$, it holds that $\|f\|_{P,1} \leq \|f\|_{Q,\beta}$.

### A.2.2 Proof for lemmas

We are ready to show the proofs now. We start with Lemma 4.

**Reminder of Lemma 4.** *Suppose $P, Q$ are two distributions supported on $\mathcal{Y}$. For every $\alpha > 1$ and $B \geq 0$, let $\beta = \frac{\alpha}{\alpha-1}$ be the Hölder conjugate of $\alpha$. The following statements are equivalent.*

- *$D_\alpha(P\|Q) \leq B$.*
- *For every function $h : \mathcal{Y} \to \mathbb{R}^{\geq 0}$, it holds that $\mathbb{E}_{y \sim P}[h(y)] \leq e^{\frac{B(\alpha-1)}{\alpha}} \mathbb{E}_{y \sim Q}[h(y)^\beta]^{1/\beta}$.*

*Proof.* First, if there is $y \in \mathcal{Y}$ such that $0 = \Pr[Q = y] < \Pr[P = y]$, then we have $D_\alpha(P\|Q) = \infty$ and Condition 2 does not hold for any $B < \infty$. In the following, we assume $\operatorname{supp}(P) = \operatorname{supp}(Q) = \mathcal{Y}$. Note that in this case, we have $D_\alpha(P\|Q) < \infty$.

We write $P(y), Q(y)$ as shorthands for $\Pr[P = y]$ and $\Pr[Q = y]$ for brevity. Now, note that $D_\alpha(P\|Q) \leq B$ is equivalent to $e^{D_\alpha(P\|Q)} \leq e^B$, which is further equivalent to

$$\mathbb{E}_{y \sim Q}\left[\frac{P(y)^\alpha}{Q(y)^\alpha}\right]^{1/\alpha} = \mathbb{E}_{y \sim P}\left[\frac{P(y)^{\alpha-1}}{Q(y)^{\alpha-1}}\right]^{1/\alpha} \leq e^{\frac{B(\alpha-1)}{\alpha}}.$$

Consider the measure space $M = (\mathcal{Y}, Q)$. By Holder's inequality, we have

$$\mathbb{E}_{y \sim Q}\left[\left(\frac{P(y)}{Q(y)}\right)^\alpha\right]^{1/\alpha} = \left\|\frac{P}{Q}\right\|_{Q,\alpha} = \sup_{h:h\not\equiv 0}\left\{\frac{\langle h, \frac{P}{Q}\rangle_Q}{\|h\|_{Q,\beta}}\right\}.$$

Moreover, since $\frac{P(y)}{Q(y)}$ is non-negative, it suffices to consider only non-negative $h$ in the supremum above. Now we are ready to verify the equivalence.

- If Condition 1 holds, we have

$$\sup_{h:h\not\equiv 0}\left\{\frac{\langle h, \frac{P}{Q}\rangle_Q}{\|h\|_{Q,\beta}}\right\} = \left\|\frac{P}{Q}\right\|_{Q,\alpha} \leq e^{\frac{B(\alpha-1)}{\alpha}}.$$

  Therefore, for every $h : \mathcal{Y} \to \mathbb{R}^{\geq 0}$, it holds that

$$\mathbb{E}_{y \sim P}[h(y)] = \mathbb{E}_{y \sim Q}\left[h(y) \cdot \frac{P(y)}{Q(y)}\right] \leq \left\|\frac{P}{Q}\right\|_{Q,\alpha} \|h\|_{Q,\beta} \leq e^{\frac{B(\alpha-1)}{\alpha}} \mathbb{E}_{y \sim Q}[h(y)^\beta]^{1/\beta}.$$

- On the other hand, if Condition 2 holds, we have

$$\left\|\frac{P}{Q}\right\|_{Q,\alpha} = \sup_{h:h\not\equiv 0}\left\{\frac{\langle h, \frac{P}{Q}\rangle_Q}{\|h\|_{Q,\beta}}\right\} \leq e^{\frac{B(\alpha-1)}{\alpha}}.$$

This completes the proof. □

The next lemma is Lemma 5.

**Reminder of Lemma 5.** *Let $\mathcal{Y}_1 \times \mathcal{Y}_2$ be a space. Consider two distributions $P, Q$ on $\mathcal{Y}_1 \times \mathcal{Y}_2$. Assume $\operatorname{supp}(P) = \operatorname{supp}(Q) = \mathcal{Y}_1 \times \mathcal{Y}_2$. Let $P_1, P_2$ be the margin of $P$ on $\mathcal{Y}_1, \mathcal{Y}_2$. For each $y_1 \in \mathcal{Y}_1$, denote by $P_2|_{P_1=y_1}$ the marginal distribution of $y_2$ conditioning on $y_1$. Also define the same notation for $Q$.*

*Let $\beta \geq 1, B \geq 0$ be two reals. For each $y_1 \in \mathcal{Y}_1$, define*

$$\ell_1(y_1) = \inf_K\left\{K : P_2|_{P_1=y_1} \preceq_\beta K \cdot Q_2|_{Q_1=y_1}\right\}.$$

*Suppose $P \preceq_\beta e^B Q$. Consider the measure spaces $(\mathcal{Y}_1, P_1(y_1) \cdot \ell_1(y_2)^{1/\beta})$ and $(\mathcal{Y}_1, Q_2)$. We have*

$$P_1 \ell_1^{1/\beta} \preceq_\beta e^B Q_1.$$

*Proof.* Suppose by contradiction that the conclusion of the lemma does not hold. That is, there is a function $g : \mathcal{Y}_1 \to \mathbb{R}^{\geq 0}$ such that

$$\|g\|_{P_1 \ell_1^{1/\beta}, 1} > \|g\|_{e^B Q_1, \beta}.$$

In the following, we show this contradicts with $P \preceq_\beta e^B Q$. First off, for each $y_1 \in \mathcal{Y}_1$, by the definition of $\ell_1(y_1)$, there is a function $f_{y_1} : \mathcal{Y}_2 \to \mathbb{R}^{\geq 0}$ such that

$$\|f_{y_1}\|_{P_2|_{P_1=y_1}, 1} = \int f_{y_1} dP_2|_{P_1=y_1} = \left( \int f_{y_1}^\beta d(\ell_1(y_1) Q_2|_{Q_1=y_1}) \right)^{1/\beta} = \|f_{y_1}\|_{\ell_1(y_1) Q_2|_{Q_1=y_1}, \beta}.$$

By scaling $f_{y_1}$ properly, we can ensure that $\|f_{y_1}\|_{P_2|_{P_1=y_1}, 1} = \ell_1(y_1)^{1/\beta}$. Consequently, we have

$$\|f_{y_1}\|_{Q_2|_{Q_1=y_1}, \beta} = \|f_{y_1}\|_{\ell_1(y_1) Q_2|_{Q_1=y_1}, \beta} \cdot \ell_1(y_1)^{-1/\beta} = 1.$$

Define a new function $f : \mathcal{Y}_1 \times \mathcal{Y}_2 \to \mathbb{R}^{\geq 0}$ as $f(y_1, y_2) = g(y_1) \cdot f_{y_1}(y_2)$. Then, we have

$$
\begin{aligned}
\|f\|_{P,1} &= \iint f(y_1, y_2) dP \\
&= \int \left( \int f_{y_1}(y_2) dP_2|_{P_1=y_1} \right) g(y_1) dP_1 \\
&= \int g(y_1) d(\ell_1^{1/\beta} P_1) \\
&> \left( \int g(y_1)^\beta d(e^B Q_1) \right)^{1/\beta} \\
&= \left( \int g(y_1)^\beta \|f_{y_1}\|_{Q_2|_{Q_1=y_1}, \beta}^\beta d(e^B Q_1) \right)^{1/\beta} \\
&= \left( \int \left( g(y_1)^\beta \int f_{y_1}(y_2)^\beta d(Q_2|_{Q_1=y_1}) \right) d(e^B Q_1) \right)^{1/\beta} \\
&= \left( \iint g(y_1)^\beta f_{y_1}(y_2)^\beta d(e^B Q) \right)^{1/\beta} \\
&= \left( \iint f^\beta d(e^B Q) \right)^{1/\beta} = \|f\|_{e^B Q, \beta}. 
\end{aligned}
\tag{29}
$$

This contradicts to the assumption that $P \preceq e^B Q$. Therefore, we conclude that such function $g$ does not exist and $P_1 \ell_1^{1/\beta} \preceq e^B Q_1$. $\qquad\square$

### A.2.3 Proof of the composition theorem

We prove the following theorem, which is equivalent to Theorem 2.

**Theorem 4.** *Let $\mathcal{M}_1, \mathcal{M}_2$ be two interactive mechanisms that run on the same data set. Suppose that $\mathcal{M}_1, \mathcal{M}_2$ are $(\alpha, \varepsilon_1), (\alpha, \varepsilon_2)$-Rényi DP, respectively. Then $\mathrm{COMP}(\mathcal{M}_1, \mathcal{M}_2)$ is $(\alpha, \varepsilon_1 + \varepsilon_2)$-Rényi DP.*

Theorem 4 implies Theorem 2 because we can interpret $\mathrm{COMP}(\mathcal{M}_1, \ldots, \mathcal{M}_k)$ as $\mathrm{COMP}(\mathrm{COMP}(\mathcal{M}_1, \ldots, \mathcal{M}_{k-1}), \mathcal{M}_k)$ and use Theorem 4 inductively. Now we prove Theorem 4.

*Proof.* Suppose without loss of generality that both mechanisms run on a single sensitive input bit $b \in \{0, 1\}$. Also suppose that there are $2T$ rounds of interactions. Starting with $\mathcal{M}_1$, the adversary communicates with two mechanisms alternately. This is without loss of generality: suppose the adversary $\mathcal{A}$ can decide the next query object based previous responses. Let $\mathbf{IT}(\mathcal{A} : \mathcal{M}_1, \mathcal{M}_2)$ be the transcript of the interaction between $\mathcal{A}$ and $\mathcal{M}_1, \mathcal{M}_2$. We reduce the interaction to a new protocol where the adversary speaks with two mechanism alternately. Let $\mathcal{A}'$ denote a modification of $\mathcal{A}$, defined as follows. $\mathcal{A}'$ simulates $\mathcal{A}$ while always alternating between two mechanisms. If the current mechanism is not the one that $\mathcal{A}$ wants to speak with, $\mathcal{A}'$ will send a special "SKIP" query, and the

mechanism responds with an "ACK" message. After this round of interaction, $\mathcal{A}'$ will switch to interact with the other mechanism, which allows it to continue simulating $\mathcal{A}$. Let $\mathbf{IT}(\mathcal{A}' : \mathcal{M}_1, \mathcal{M}_2)$ denotes the transcript of the new interaction. Therefore, for $b \in \{0, 1\}$, it is easy to establish a bijection between $\mathrm{supp}(\mathbf{IT}(\mathcal{A}' : \mathcal{M}_1^b, \mathcal{M}_2^b))$ and $\mathrm{supp}(\mathbf{IT}(\mathcal{A} : \mathcal{M}_1^b, \mathcal{M}_2^b))$. Moreover, the bijection mapping is independent of $b$[4]. Therefore, bounding the divergences between $\mathbf{IT}(\mathcal{A} : \mathcal{M}_1^b, \mathcal{M}_2^b)$, $b \in \{0, 1\}$ is equivalent to bounding those between $\mathbf{IT}(\mathcal{A}' : \mathcal{M}_1^b, \mathcal{M}_2^b)$, $b \in \{0, 1\}$.

Let $\mathcal{Y}, \mathcal{Z}$ denote the response domains of $\mathcal{M}_1, \mathcal{M}_2$ respectively. Also let $y_1, \ldots, y_T$, $z_1, \ldots, z_T$ denote the lists of responses returned by $\mathcal{M}_1$ and $\mathcal{M}_2$ respectively. We assume that each response $y_i, z_j$ contains a copy of the corresponding query message (so that we can recover the whole interaction history just from the responses).

Now, fix $\mathcal{A}$ to be an arbitrary adversary. Let $P, Q \in \Delta((\mathcal{Y} \times \mathcal{Z})^T)$ denote the output distributions when $\mathcal{A}$ interacts with $(\mathcal{M}_1^0, \mathcal{M}_2^0)$ and $(\mathcal{M}_1^1, \mathcal{M}_2^1)$ respectively. Our goal is to prove that

$$\max \{D_\alpha(P\|Q), D_\alpha(Q\|P)\} \leq \varepsilon_1 + \varepsilon_2.$$

We bound $D_\alpha(P\|Q)$ below. The bound for $D_\alpha(Q\|P)$ is symmetric.

For a distribution $D$, we always use $D(x)$ to denote $\Pr[D = x]$. Write $y = (y_1, \ldots, y_T)$ where $y_i$ denotes the $i$-th response. Also write $z = (z_1, \ldots, z_T)$ and denote $yz := (y_1, z_1, \ldots, y_T, z_T)$. By Lemma 4, it suffices to show that for every $h \colon (\mathcal{Y} \times \mathcal{Z})^T \to \mathbb{R}^{\geq 0}$, it holds that

$$\sum_{y \in \mathcal{Y}^T, z \in \mathcal{Z}^T} P(yz)h(yz) \leq \left( e^{\varepsilon_1 + \varepsilon_2} \sum_{y \in \mathcal{Y}^T, z \in \mathcal{Z}^T} Q(yz)h(yz)^\beta \right)^{1/\beta} \tag{30}$$

where $\beta = \frac{\alpha}{\alpha-1}$ is the Hölder conjugate of $\alpha$.

For each $i \in [T]$, let $P_i^y, P_i^z$ be the projection of $P$ onto $y_i, z_i$. For each $i \in [T]$, let $y_{\leq i}, z_{\leq i}$ denote the first $i$ responses from $y$ and $z$. Denote $(yz)_{\leq i} = (y_1, z_1, \ldots, y_i, z_i)$. Then, let $P_i^y|_{yz_{<i}}$ denote the distribution of $y_i$ conditioning on $(yz)_{<i}$, and $P_i^z|_{yz_{<i}, y_i}$ denote the distribution of $z_i$ conditioning on $(yz)_{<i}$ and $y_i$. Also define the same notation for $Q$. Then we write

$$\sum_{y \in \mathcal{Y}^T, z \in \mathcal{Z}^T} P(yz)h(yz) = \sum_{(yz)_{\leq T-1}} P((yz)_{\leq T-1}) \sum_{y_T, z_T} P_T^y|_{yz_{<T}}(y_T) P_T^z|_{yz_{<T}, y_T}(z_T) h(yz) \tag{31}$$

For every $t < T$ and every $y_{\leq t}$, let $\mathcal{M}_1^0|_{y_{\leq t}}$ (resp. $\mathcal{M}_1^1|_{y_{\leq t}}$) denote the interactive system $\mathcal{M}_1^0$ (resp. $\mathcal{M}_1^1$) conditioning on that it has answered $y_1, \ldots, y_t$ to the first $t$ queries. Formally, for every $(x_{t+1}, y_{t+1}), \ldots, (x_{t'}, y_{t'})$ and $x_{t'+1}$, define

$$\mathcal{M}_1^b|_{y_{\leq t}}((x_i, y_i)_{t < i \leq t'}, x_{t'+1}) := \mathcal{M}_1^b((x_i, y_i)_{1 \leq i \leq t'}, x_{t'+1}).$$

We also define the same notation for the second mechanism $\mathcal{M}_2$. Next, define

$$\ell_t(y_{\leq t}) := \exp \left( \sup_{A:\text{adversary}} \left\{ D_\alpha\left(\mathbf{IT}(A : \mathcal{M}_1^0|_{y_{\leq t}}) \| \mathbf{IT}(A : \mathcal{M}_1^1|_{y_{\leq t}})\right) \right\} \right) \tag{32}$$

and

$$r_t(z_{\leq t}) := \exp \left( \sup_{A:\text{adversary}} \left\{ D_\alpha\left(\mathbf{IT}(A : \mathcal{M}_2^0|_{z_{\leq t}}) \| \mathbf{IT}(A : \mathcal{M}_2^1|_{z_{\leq t}})\right) \right\} \right). \tag{33}$$

By the assumed Rényi DP guarantee, we have that $\ell_0(\emptyset) \leq e^{\varepsilon_1}$ and $r_0(\emptyset) \leq e^{\varepsilon_2}$. We claim the following.

**Claim 3.** *For each $t \leq T - 1$ and $y_{<t}, z_{<t}$, consider two measures $P_t^y|_{yz_{<t}}(y)\ell_t(y_{<t} \circ y)$ and $Q_t^y|_{yz_{<t}}(y)$ on the space $\mathcal{Y}$ (here $\circ$ denotes concatenation). It holds that*

$$P_t^y|_{yz_{<t}}(y)\ell_t(y_{<t} \circ y)^{1/\beta} \preceq_\beta \ell_{t-1}(y_{<t}) Q_t^y|_{yz_{<t}}(y).$$

*A symmetric conclusion holds for $P^z$ and $z$. Namely*

$$P_t^z|_{yz_{<t}, y_t}(z)r_t(z_{<t} \circ z)^{1/\beta} \preceq_\beta r_{t-1}(z_{<t}) Q_t^z|_{yz_{<t}, y_t}(z).$$

---

[4]This is to say, suppose $\mathcal{M}_1^0, \mathcal{M}_2^0, \mathcal{M}_1^1, \mathcal{M}_2^1$ are four systems, then the bijection between $\mathrm{supp}(\mathbf{IT}(\mathcal{A} : \mathcal{M}_1^b, \mathcal{M}_2^b))$ and $\mathrm{supp}(\mathbf{IT}(\mathcal{A}' : \mathcal{M}_1^b, \mathcal{M}_2^b))$ would be the same for $b \in \{0, 1\}$.

*Proof.* Construct an adversary $\mathcal{A}'$ interacting with $\mathcal{M}_1^b|_{y_{<t}}$ as follows. $\mathcal{A}'$ starts $\mathcal{A}$ with the conditioning that $\mathcal{A}$ has gone through the interaction history $yz_{<t}$. Then $\mathcal{A}'$ simulates one step of $\mathcal{A}$ and sends a query to $\mathcal{M}_1^b|_{y_{<t}}$. Upon receiving the response $y$, $\mathcal{A}'$ observes $y_t$ and switches to run the optimal adversary against $\mathcal{M}_1^b|_{y_{\leq t}}$ provided by (32). By definition, $\mathcal{M}_1^b|_{y_{\leq t}}$ is $(\alpha, \log(\ell_{t-1}(y_{<t})))$-Rényi DP. Applying Lemma 5 on $\mathbf{IT}(\mathcal{A}', \mathcal{M}_1^b|_{y_{<t}})$ completes the proof. The proof for $P_t^z$ is similar. $\qquad\square$

Turning back to (31), we first deduce that

$$\sum_{(yz)_{\leq T-1}} P((yz)_{\leq T-1}) \sum_{y_T, z_T} P_T^y|_{yz_{<T}}(y_T) P_T^z|_{yz_{<T},y_T}(z_T) h(yz)$$

$$\leq \sum_{(yz)_{\leq T-1}} P((yz)_{\leq T-1}) \sum_{y_T} P_T^y|_{yz_{<T}}(y_T) \left( r_{T-1}(z_{<T}) \sum_{z_T} Q_T^z|_{yz_{<T},y_T}(z_T) h(yz)^\beta \right)^{1/\beta}$$

$$\leq \sum_{(yz)_{\leq T-1}} P((yz)_{\leq T-1}) \left( r_{T-1}(z_{<T}) \ell_{T-1}(y_{<T}) \sum_{y_T,z_T} Q_T^y|_{yz_{<T}}(y_T)\, Q_T^z|_{yz_{<T},y_T}(z_T) h(yz)^\beta \right)^{1/\beta}.$$
(34)

So far we haven't utilized Claim 3 yet. Denote

$$H(yz_{\leq T-1}) := \left( \ell_{T-1}(y_{<T}) \sum_{y_T,z_T} Q_T^y|_{yz_{<T}}(y_T)\, Q_T^z|_{yz_{<T},y_T}(z_T) h(yz)^\beta \right)^{1/\beta}.$$

Applying Claim 3 on (34) for $P_{T-1}^z$ yields that

$$\sum_{(yz)_{\leq T-2}, y_{T-1}} P((yz)_{\leq T-2}, y_{T-1}) \sum_{z_{T-1}} P_{T-1}^z|_{yz_{\leq T-2},y_{T-1}}(z_{T-1}) r_{T-1}(z_{<T})^{1/\beta} H$$

$$\leq \sum_{(yz)_{\leq T-2}, y_{T-1}} P((yz)_{\leq T-2}, y_{T-1}) \left( r_{T-2}(z_{\leq T-2}) \sum_{z_{T-1}} Q_{T-1}^z|_{yz_{\leq T-2},y_{T-1}}(z_{T-1}) H^\beta \right)^{1/\beta}.$$
(35)

We proceed to apply Claim 3 on (35) for $P_{T-1}^y, P_{T-2}^z, P_{T-2}^y \ldots, P_1^z, P_1^y$ in order. We can get

$$\sum_{(yz)_{\leq T-1}} P((yz)_{\leq T-1}) \sum_{y_T, z_T} P_T^y|_{yz_{<T}}(y_T) P_T^z|_{yz_{<T},y_T}(z_T) h(yz)$$

$$\leq \left( \ell_0(\emptyset) r_0(\emptyset) \sum_{yz} Q(yz) h(yz)^\beta \right)^{1/\beta}.$$
(36)

This shows that $P \preceq e^{\varepsilon_1 + \varepsilon_2} Q$, which consequently implies that $D_\alpha(P\|Q) \leq \varepsilon_1 + \varepsilon_2$. Similarly, we can bound $D_\alpha(Q\|P) \leq \varepsilon_1 + \varepsilon_2$. Combining two bounds together completes the proof. $\qquad\square$

### A.3 Proof for Concentrated DP

In this section, we prove Corollary 1. We recall the definition of zero-concentrated DP and truncated concentrated DP.

**Definition 4** (zero-concentrated differential privacy, Bun and Steinke [2016]). *Let $\rho > 0$ be a real and $\mathcal{M}$ be a mechanism. $\mathcal{M}$ is called $\rho$-zero-concentrated DP (or $\rho$-zCDP for short), if for every $\alpha \in (1, +\infty)$, $\mathcal{M}$ is $(\alpha, \alpha \cdot \rho)$-RDP.*

**Definition 5** (truncated concentrated differential privacy Bun et al. [2018]). *Let $\rho > 0, \omega > 1$ be two reals, and $\mathcal{M}$ be a mechanism. $\mathcal{M}$ is called $(\rho, \omega)$-truncated DP (or $(\rho, \omega)$-tCDP), if for every $\alpha \in (1, \omega)$, $\mathcal{M}$ is $(\alpha, \alpha \cdot \rho)$-RDP.*

We are ready to prove Corollary 1 below.

*Proof.* We first prove for zCDP. Suppose $\mathcal{M}_1, \ldots, \mathcal{M}_k$ are $k$ interactive mechanisms, where for each $i \in [k]$, $\mathcal{M}_i$ is $\rho_i$-zCDP. By definition, we know that $\mathcal{M}_i$ is $(\alpha, \alpha\rho_i)$-RDP for every $\alpha > 1$. By Theorem 2, we know that $\text{COMP}(\mathcal{M}_1, \ldots, \mathcal{M}_k)$ satisfies $(\alpha, \alpha(\sum_i \rho_i))$-RDP. Since this argument holds for every $\alpha > 1$, we conclude that $\text{COMP}(\mathcal{M}_1, \ldots, \mathcal{M}_k)$ satisfies $(\sum_i \rho_i)$-zCDP.

The proof for tCDP is similar. Fix $\omega > 1$. Again let $\mathcal{M}_1, \ldots, \mathcal{M}_k$ are $k$ interactive mechanisms, where for each $i \in [k]$, $\mathcal{M}_i$ is $(\rho_i, \omega)$-tCDP. Then, for every $\alpha \in (1, \omega)$, we know that $\mathcal{M}_i$ is $(\alpha, \alpha\rho_i)$-RDP by definition. Theorem 2 then shows that $\text{COMP}(\mathcal{M}_1, \ldots, \mathcal{M}_k)$ satisfies $(\alpha, \alpha(\sum_i \rho_i))$-RDP. Since the argument holds for every $\alpha \in (1, \omega)$, we conclude that $\text{COMP}(\mathcal{M}_1, \ldots, \mathcal{M}_k)$ satisfies $(\sum_i \rho_i, \omega)$-tCDP. □

## B  A Motivating Example of Concurrent Composition

To demonstrate the power of concurrent composition, in this section, we use Theorem 1 to analyze a simple private "Guess-and-Check" algorithm. We remark that this is a rather preliminary application: the weaker concurrent composition theorem by Vadhan and Wang [2021] is sufficient to do the job. However, the main purpose of this section is to highlight the importance of concurrent composition, and hopefully inspire researchers to design more sophisticated algorithms.

**Setup.** Now we describe the problem. The private algorithm holds a sensitive data set $X$. The user keeps issuing queries to the algorithm, where each query consists of a 1-Lipschitz function $f_i$ and a guess $\tau_i \in \mathbb{R}$ for the value of $f_i(X)$. The algorithm's job is to verify if $f_i(X) \approx \tau_i$. If it is the case, the algorithm reports "PASS" and continues to the next query. Otherwise, the algorithm reports "WRONG" and a value $v_i$ that is approximately equal to $f_i(X)$ (i.e., the algorithm not only declares the invalidity of the user's guess, but also provides a correct estimation for $f_i(X)$).

We consider the following algorithm.

---
**Algorithm 1:** The Private Guess-and-Check

**Input:** Private dataset $X$. Error tolerance parameter $E > 0$. Privacy-related parameters $c \geq 1, \varepsilon \in (0, 1)$.

1 **Program:**
2     $\rho \leftarrow \text{Lap}\left(\frac{1}{\varepsilon}\right)$      `// Note that this noise has standard deviation ` $\approx \frac{1}{\varepsilon}$
3     **for** $i = 1, 2, \ldots,$ **do**
4         Receive the next query $(f_i, \tau_i)$
5         $\gamma_i \leftarrow \text{Lap}(c/\varepsilon)$
6         **if** $|f_i(X) - \tau_i| + \gamma_i \geq E + \rho$ **then**
7             $v_i \leftarrow f_i(X) + \text{Lap}(c/\varepsilon)$
8             Report $(\text{WRONG}, v_i)$
9             $t \leftarrow t + 1$
10             **if** $t = c$ **then**
11                 HALT the algorithm.
12         **else**
13             Return PASS

---

**Discussions.** Algorithm 1 is parameterized by an error tolerance parameter $E > 0$ and two privacy parameters $c \geq 1, \varepsilon \in (0, 1)$. Roughly speaking, it can process queries until identifying at least $c$ queries whose guesses deviate from the true value by at least (roughly) $E$. It works by (concurrently) composing a variant of the sparse vector technique by Lyu et al. [2017] with the standard Laplace noise-adding mechanism.

The main advantage of the Lyu et al. [2017] SVT is that it only adds noise to the threshold once (Line 2 of algorithm 1), using a *much smaller* noise, which makes the SVT algorithm more accurate. Since the utility guarantee of the algorithm is not the focus of this work, we omit more discussions here and refer interested readers to [Lyu et al., 2017, Zhu and Wang, 2020] for more detail.

We consider the privacy guarantee of Algorithm 1. In fact, without the concurrent composition framework, it is not clear whether or not Algorithm 1 is really private! If we replace Line 7 of

the algorithm by $v_i \leftarrow 0$, then the algorithm is indeed $(3\varepsilon, 0)$-private, because it is just a faithful implementation of the Lyu et al. [2017] SVT. However, in Algorithm 1, the algorithm reports a correct estimation $v_i$ for each inaccurate guess, which implies that the future query to the algorithm may depend on $v_i$, and thus on the private data set $X$. In this case, the original analysis from [Lyu et al., 2017] does not hold anymore.

**Analyzing the privacy.** While it is not hard to prove the privacy property of Algorithm 1 by examining the proof of Lyu et al. [2017] carefully and applying some modifications, here we show that Algorithm 1 admits a fairly straightforward privacy proof under the concurrent composition framework, using the privacy theorem by Lyu et al. [2017] as a black box. We do the analysis now. First, we have the following lemma from [Lyu et al., 2017].

**Lemma 6** (Theorem 2 in Lyu et al. [2017]). *Consider replacing Line 7 of Algorithm 1 with $v_i \leftarrow 0$. The resulting algorithm is $(3\varepsilon, 0)$-DP.*

The following fact is well known.

**Lemma 7** (Laplace mechanism). *Consider the following algorithm: given a list of $c$ adaptively chosen, 1-Lipschitz queries $(g_1, \cdots, g_c)$, answer each query with $g_i(X) + \mathrm{Lap}(c/\varepsilon)$. The algorithm is $(\varepsilon, 0)$-DP.*

Combining Lemmas 6 and 7 under the concurrent composition framework directly yields the following result.

**Theorem 5.** *Algorithm 1 is $(4\varepsilon, 0)$-DP.*

*Proof.* Consider simulating Algorithm 1 by concurrently composing two algorithms $A_1, A_2$. $A_1$ is just a modification of Algorithm 1 where we replace Line 7 in Algorithm 1 with $v_i \leftarrow 0$. By Lemma 6, $A_1$ is $(3\varepsilon, 0)$-DP. $A_2$ accepts at most $c$ 1-Lipschitz query. For each query $g_i$, $A_2$ responds with $g_i(X) + \mathrm{Lap}(c/\varepsilon)$. By Lemma 7, $A_2$ is $(\varepsilon, 0)$-DP. By Theorem 1, $\mathrm{COMP}(A_1, A_2)$ is $(4\varepsilon, 0)$-DP.

We now describe how to simulate Algorithm 1 with $\mathrm{COMP}(A_1, A_2)$. For each query $(f_i, \tau_i)$ to Algorithm 1, we first feed it into $A_1$ and observe the outcome. We pass this query if the outcome is PASS. Otherwise, the outcome must be (WRONG, 0). We then query $A_2$ with $f_i$ to get an estimation $f_i + \mathrm{Lap}(c/\varepsilon)$, and think of this estimation as the "$v_i$" returned by Algorithm 1. In this way, it is easy to see that we faithfully simulate Algorithm 1 by interacting with $\mathrm{COMP}(A_1, A_2)$. Since $\mathrm{COMP}(A_1, A_2)$ is $(4\varepsilon, 0)$-DP, Algorithm 1 must be $(4\varepsilon, 0)$-DP also. This completes the proof. □

**Remark 2.** *Finally, we remark that a similar private "Guess-and-Check" algorithm was also proposed and analyzed by Zhu and Wang [2020], where the authors also considered using a version of SVT* without *refreshing the threshold after answering each "meaningful" query. Therefore, their algorithm is also subject to the concurrent composition issue, which seems to be overlooked in the original analysis of Zhu and Wang [2020]. Since they were working with Rényi DP, our Theorem 2 provides a remedy to this issue easily.*