# OpenReview forum: "Composition Theorems for Interactive Differential Privacy"
_NeurIPS.cc/2022/Conference — NeurIPS 2022 Accept_

### Official Review · Reviewer_PTTm · 2022-07-07

**Rating:** 7
**Confidence:** 3
**Soundness:** 3 good
**Presentation:** 3 good
**Contribution:** 3 good

**Summary:**

This paper deals with exploring the “cost” of a concurrent composition of *interactive* DP mechanisms, compared to the (standard) non-interactive case.

The well-known composition theorems for DP mechanisms only hold when the mechanisms are non-interactive.
However, not all known mechanisms are non-interactive. The best example is the Sparse-Vector Technique, which is a very popular interactive mechanism, where the adversary in each round can (adaptively) choose a new query.
A natural question that arises is whether we can generalize the well-known composition theorems for non-interactive mechanisms to the interactive case in which several mechanisms can be executed concurrently (or alternatively, does it matter if we compose them sequentially or concurrently in terms of the obtained privacy guarantee?)

It is important to note that in other settings, the answer to similar results is not always positive. For instance, in the context of interactive proof systems, it is well-known that parallel-repetition does not increase soundness as sequential repetitions for 2-prover (and multi-prover) games and for interactive arguments (aka, computationally sound proofs).

However, in the context of interactive DP mechanisms, the situation seems more positive.

Vadhan and Wang 21 initiate the study of concurrent composition of interactive mechanisms. They showed that composing pure DP interactive mechanisms has the same cost as composing non-interactive mechanisms. However, for approximate DP mechanisms (i.e., with a parameter $\delta>0$), they only provided a very basic (and wasteful) composition theorem.

This work makes two major steps towards a positive answer: First, they extend the tight technique of Vadhan and Wang 2021 in the pure DP case for the approximate DP case, yielding that the cost of concurrent composition of approximate DP interactive mechanisms is the same as in the non-interactive case. Second, they provide a similar result (using a different technique) for Renyi DP, which in particular implies such results for other popular notions of privacy like zCDP and tCDP.


**Questions:**

High-Level Questions:
1) I’m wondering if each privacy loss measure (e.g., $(\varepsilon,\delta)$-DP, Renyi-DP, f-DP) requires a unique and tailored proof technique, or if one can find more general high-level arguments for showing equivalences between sequential and concurrent composition of interactive mechanisms.
Can you explain (in high level) why the proof technique for $(\varepsilon,\delta)$-DP does not hold for Renyi-DP (and vice versa)?
2) What about other approximate notions of differential privacy, i.e., approximate zCDP which is also very useful. Can you provide tight composition theorems for it by somehow combining your two results?


Technical Question:
1) In the supplementary material, lines 548-551, there are four cases, and you claim that “It is easy to see that if none of the above happens, we can keep increasing $tE^{0/1}$ “. Can you elaborate? E.g., what happens if $tE^0$ is tight at $y_{t+1}$ but $tE^1$ is not? Why $Gap_1$ must be zero in that case?

Minor issues/typos:
1) Line 16: [Dwork et al. 2016] ->  [Dwork et al. 2006]
2) Line 20: A remains -> M remains
3) Line 90: $(\alpha,\varepsilon)$ -> $(\varepsilon,\delta)$
4) I suggest to write $(\varepsilon,\delta)$-DP instead of writing $(\varepsilon,\delta)$-approximate DP all the time. This is very standard. You can mention it in Definition 2.
5) Line 180: $IT(N^b:A)$ should be $IT(A:N^b)$ (be consistent).
6) Line 213, the right term: missing $\in S$ in both terms.
7) Paragraph 216-223: Presents a different proof technique than what is written in the supplementary material.
8) “The bound for $D_{\alpha}(P||Q)$” -> “The bound for $D_{\alpha}(Q||P)$”
9) Line 324: $Z_T$ -> $z_T$

Minor issues/typos in the supplementary material:
1) Equation 18 (and other places): What is the difference between $E^b$ and $\mathcal{E}^b$?
2) Line 493: the -> then
3) Line 497: contron -> control
4) Equation 21: Why don’t you use the notation of $M^b((y_i),(x_i))$?
5) Equation 22 and many other places: Write max{*} instead of max(*) to make it more readable.
6) Equation after line 513: No second term in the max.
7) I suggest replacing $tE$ with $\tilde{E}$.
8) Line 598: $B_1,\ldots,B$ $->$ $B_1,\ldots,B_k$.
9) Equation after line 603, right term: $M_1^b -> M_{1,j}^b$ and  $M_k^b -> M_{k,j}^b$.


**Limitations:**

In the “Conclusion and Future Directions” section (section 4), the authors address the limitations of their results.

**Strengths And Weaknesses:**

Strengths:
1) The results are important and impressive.
2) The proof techniques of both cases (DP and Renyi-DP) are interesting.
3) Overall, the paper is well-organized, and looks sound - All proofs are provided in the supplementary material. I read the full proof for the DP case and I didn’t find something false (There were a few things that I didn’t understand, see my questions below).

Weaknesses:
1) I find the two different proof techniques for $(\varepsilon,\delta)$-DP and Renyi-DP as a disadvantage rather than an advantage. It is unclear to me if there is something inherently different between the privacy loss measures in this context of sequential vs concurrent composition.
2) While the authors did an effort to give an intuition for the proof in the Renyi-DP case, I still couldn’t understand the idea behind all the calculations.
3) There are many typos and other minor issues in the writing (see examples in the next paragraph).

Overall, I think the strengths of this paper are more significant than its weaknesses, so I recommend accepting it.

---

> ### Author Response · Authors · 2022-07-28
> **Our response to Reviewer PTTm**
>
>
> Thanks for your kind and thorough review! Here is our response to your questions.
>
> **High-level questions:**
>
> * Given our results, finding a unified approach to prove concurrent composition theorems for different variants of DP is a natural and interesting question. Here we try to describe (in a high level) why our current proof needs two different approaches to handle $(\varepsilon,\delta)$-DP and Renyi DP. Roughly speaking, the "main challenges" in the two proofs are very different:
>
>   * In $(\varepsilon,\delta)$-DP, the main difficulty is to deal with the $\delta$ probability of failure. Namely, the mechanism can be blatantly non-private with some small but non-zero probability. Capturing this $\delta$-probability of "bad behavior" turns out to be quite challenging.
>   * In Renyi-DP, the main difficulty is to deal with "non-linearity" of the privacy definition. While Renyi divergence satisfies many natural and nice properties (such as post-processing and sequential composition), it fails to satisfy some other desired properties due to the non-linearity of the definition (such as convexity, see (https://arxiv.org/pdf/1206.2459.pdf) for a counterexample).
>
>   Having said that, we do not view the two proofs as two unrelated arguments. Indeed, we believe there is a way to connect the two proofs to arrive at a unified proof for concurrent composition. This is also connected to your second question: see my response below.
>
> * This is a very good question. We believe the optimal composition theorem for approximate zCDP holds. It might be possible to prove it by combining our two proofs: for example, we can try to use the idea from our first proof (for $(\varepsilon,\delta)$-DP) to show that one can decompose the pair of neighboring systems into "well-behaved" parts and "bad" parts, where the pair of "well-behaved" parts are subject to RDP constraint, allowing us to argue the privacy by using the idea from our second proof (for RDP). This is a natural question for future research.
>
> **Technical question:**
>
> Let us recall the setting (simplified for better illustration). We have defined $Gap_{b}$ as:
> $$
> Gap_b:= C^b - \sum_{y_{t+1}} tE^b(y_{t+1}), ~~~\forall b\in \{0,1\}
> $$
> Also, Equation (26) requires that:
> $$
> \delta \cdot \mathrm{tE}^b(y_{t+1}) \le D^b(y_{t+1}) + e^{-\varepsilon} \delta \cdot \mathrm{tE}^{1-b}(y_{t+1}), \forall b\in \{0,1\}, y_{t+1}
> $$
> Here, $C^b, D^b(y_{t+1})$ are fixed values. Our goal is to decrease $Gap_{0/1}$ to zero by increasing $\mathrm{tE}^{0/1}(y_{t+1})$, without compromising Equation (26). Our strategy is to increase $\mathrm{tE}^{0/1}(y_{t+1})$ as far as we can. Then, we observe that there might be two reasons that we cannot increase $\mathrm{tE}^{b}(y_{t+1})$:
>
> * $\mathrm{Gap}_b = 0$. In this case, there is no need to further increase $\mathrm{tE}^{b}$.
> * $\mathrm{tE}^b$ is tight at $y_{t+1}$. In this case, further increasing $\mathrm{tE}^{b}(y_{t+1})$ will violate Equation (26) at $(b,y_{t+1})$.
>
> Back to your example, if $\mathrm{tE}^1$ is not tight at $y_{t+1}$, then the only reason that we cannot increase $\mathrm{tE}^1(y_{t+1})$ is that $\mathrm{Gap}_1 = 0$.
>
> We will update this illustration in the final version of the paper.
>
> **Other comments**:
>
> (1): Thanks for pointing this out.
>
> (4): Thanks for the suggestion. We will update the definitions accordingly.
>
> (2, 3, 5-9): fixed. Thanks!
>
> **In supplementary material:**
>
> (1) We use $\mathcal{E}$ to denote the system (i.e., the randomized algorithm $\mathcal{E}:(X\times Y)^*\times X\to Y$), and use $E$ to denote the "PDF" or $\mathcal{E}$. e.g., $E(y_{t}) = \Pr[\mathcal{E} \text{ coutputs } (y_t) ]$.
>
> (7) Thanks for the nice suggestion. We will update the proof accordingly.
>
> (2-6, 8, 9) fixed. Thanks!
>
> I hope this answers your questions! Please let me know if you have any further questions.

---

> > ### Comment · Reviewer_PTTm · 2022-08-07
> > **response**
> >
> > No further questions to the authors.
> >
> > In my opinion, this is an impressive paper that tackles an important and challenging problem.
> > The writing can be improved (based on the comments of all the reviewers), and I believe the authors will do the effort to do that.
> > Therefore, unless someone finds a bug in the proofs, I recommend accepting the paper.

---

### Official Review · Reviewer_uLrt · 2022-07-09

**Rating:** 6
**Confidence:** 3
**Soundness:** 3 good
**Presentation:** 3 good
**Contribution:** 3 good

**Summary:**

This work studies the problem of concurrent composition for differential privacy. The authors provide formal proofs for differential privacy guarantees for parallel composition for different notions of differential privacy. First, they extend the results by Vadhan and Wang [2021] on concurrent compositions with pure differential privacy, then they extended those results to also include other privacy notions such as Renyi DP and zero-concentrated DP.

**Questions:**

If possible, it would be nice to discuss how the adversary can use the query history to gain more information from the current query.

**Limitations:**

The authors discuss the limitations of the work, where possible future directions and extensions are discussed.

**Strengths And Weaknesses:**

The paper studies a significant problem, and the results are original. Extending the results for sequential composition theorems to concurrent theorems is an important step towards better understanding privacy. This work is interesting towards at least some audience.

The paper is well written and clear, however, some issues need to be clarified. The paper needs some proofreading still, but next are some examples of things to double check:

-The distinction between mechanisms and systems is mentioned in section 1.1, however, in laters parts of the paper, it seems that those are used interchangeably. For instance, on page 2, "Suppose $M_1,M_2,\cdots, M_k$ are $k$ systems." As from the definition above, should those be mechanisms instead?

- In section 1.2, "that $IT(A: M_1^0,\cdots,M_k^0)$ and $IT(A: M_1^0,\cdots,M_k^0)$, shouldn't the second one have superscript $1$?

---

> ### Author Response · Authors · 2022-07-28
> **Our response to Reviewer uLrt**
>
> We thank Reviewer uLrt for the kind review and questions.
>
> * On page 2, we do mean to use "systems", because in the paragraph from Line 75-84, we are defining the composition of k interactive systems in general. In particular, we do not care whether or not these M_i's are from running a mechanism on a private data set $d$.
>
>   In most part of the paper, we are studying interactive systems that are induced by running mechanisms on a pair of neighboring data sets. However, we find that referring to the two systems as $M^0$, $M^1$ less verbose than repeatedly saying "the mechanism $M$ running on the data set $d$ and $d'$". We will double-check the paper to make sure we are using the two terminologies consistently.
>
> * Yes, the second term should have superscript $1$. Thanks for pointing out the typo.
>
> About the last question:
>
> * This is a very good question. Thanks for asking! First of all, since we have shown that the optimal composition theorem holds for the concurrent composition, the adversary cannot really gain any advantage from the query history. However, this is far from obvious without a formal proof.
> * For example, one could have an intuition as follows: when the adversary is interacting with a private mechanism $M_1$, there are many queries they can choose, and it is possible that some queries will cause $M_1$ to "leak" more information. Without any extra clue, the adversary can only choose a query from all possible queries "randomly", and the privacy of $M_1$ is warranted in this case. However, when we consider the concurrent composition of $M_1$ with another independently running mechanism $M_2$, it might be possible that the adversary can get some "hint" from their interaction $M_2$. This "hint" can in turn help the adversary to ask the "correct" question to $M_1$, thereby causing $M_1$ to compromise more privacy. Moreover, when the adversary interleaves its queries to $M_1,M_2$, it might be possible to "boost" the effectiveness of the attack, so that the overall privacy warranted by $COMP(M_1,M_2)$ is significantly worse than the setting of sequential composition.
> * If one does not know whether the optimal concurrent composition theorem holds, the intuition sketched above could be very appealing. Fortunately, our result strongly refutes this intuition by confirming the correctness of the optimal concurrent composition theorem.
>
> I hope this answers your question. In the final version of the paper, we will address the typo and the confusing point you mentioned, as well as discuss the implications of our result. Please let us know if you have any further questions.

---

> > ### Comment · Reviewer_uLrt · 2022-08-08
> > **Thank you for the response**
> >
> > Thank you for clairifying my concerns. I have no further questions for the authors.

---

### Official Review · Reviewer_2XK9 · 2022-07-15

**Rating:** 3
**Confidence:** 3
**Soundness:** 2 fair
**Presentation:** 2 fair
**Contribution:** 2 fair

**Summary:**

The paper, as the title suggests, studies optimal composition theorems for interactive differential privacy, following the framework of [Vadhan and Wang 2021] and resolving one of the open questions in that paper.


**Questions:**

See below

**Limitations:**

Unfortunately, I still find the paper missing an important piece, which is an explicit example illustrating how concurrent composition offers more than the existing framework ([Mironov, 2017] and [Dong et al., 2022]. Are there algorithms that previous works cannot analyze tightly or previous analysis is wrong/faulty? I also couldn't find anything helpful in [Vadhan and Wang 2021].

The framework seems novel indeed. However, if I were to do it, I would break each "system" $M_i$ into smaller pieces. Namely, as in Definition 1, each $M_i$ is a randomized mapping $Y^*\to Y$ and hence can be decomposed into a sequence of traditional mechanisms: $M_{ij}:Y^j\to Y,j=0,1,2,\ldots$. If each of these $M_{ij}$ is private, then one can compose using the existing results even when they interleave (say the adversary decides to use $M_{i_t,t}$ after observing $t$ outcomes). What does interactive DP offer exactly? I would like to see an explicit playground where this question can be answered clearly.


**Strengths And Weaknesses:**

I like the simplified definitions of interactive system and concurrent composition, compared to [Vadhan and Wang 2021].

---

> ### Author Response · Authors · 2022-07-28
> **Our response**
>
>
> We thank Reviewer 2XK9 for the review and the question. Here is our response to your question:
>
> Perhaps the best example to illustrate the power of concurrent compositions is the Sparse Vector Technique. Suppose $M_1$ is an instantiation of an $(\varepsilon, 0)$-DP sparse vector (a.k.a. AboveThreshold) algorithm. Then it is impossible to decompose $M_1$ into a list of private pieces $M_{1,1},M_{1,2},\dots, M_{1,T}$. There are two obvious reasons: (1) The responses given by $M_{1,1},\dots, M_{1,T}$'s are correlated, due to the fact that the AboveThreshold algorithm needs to add a noise to the threshold during the initialization, and all of $M_{1,i}$'s are using the same noisy threshold. (2) The initialization step is crucial for the privacy proof of Sparse Vector Technique. Since $M_{1,1},\dots, M_{1,T}$ share the same noisy threshold, it is hard to say that they are individually private.
>
> Even if one does find such a decomposition, and manages to show that each $M_{1,i}$ is $(\varepsilon,0)$-DP, the privacy guarantee warranted by the sequential composition can be significantly worse. When the adversary runs $M_1$ concurrently with another interactive mechanism, say, an $(\varepsilon, 0)$-DP algorithm $M_2$, the overall privacy guarantee given by the sequential composition is only $(T\varepsilon, 0)$-DP or $(4\varepsilon\sqrt{T \log(1/\delta)}, \delta)$-DP, because we have decomposed the $(\varepsilon, 0)$-DP interactive mechanism $M_1$ into a list of $T$ $(\varepsilon, 0)$ non-interactive mechanisms $M_{1,1},\dots, M_{1,T}$. In contrast, using our concurrent composition theorem (or the Vadhan-Wang theorem, since we start from pure-DP in this case) shows that the resulting composition is $(2\varepsilon, 0)$-DP.
>
> To summarize, if one assumes that the interactive mechanism $M_1$ admits a decomposition into a list of (1) private, (2) non-interactive, and (3) independent mechanisms $M_{1,1},M_{1,2},\dots, M_{1,T}$, then it is possible to use the standard composition theorem to argue the privacy of the concurrently composited mechanism. However, these assumptions are arguably restrictive: they exclude some of the most powerful private algorithms, such as Sparse Vector Technique and Private Multiplicative Weight Update. The point of [Vadhan-Wang'21] and our work is that we have no assumption on $M_1$: the only requirement is that the $T$-round interaction history between $M_1$ and the adversary is private.
>
> I hope this answers your question. Please let us know if you have any further questions.

---

> > ### Comment · Reviewer_2XK9 · 2022-08-07
> > **TIME SENSITIVE: Please revise the paper**
> >
> > Dear authors,
> >
> > I regret to not having realized before yesterday that next week's discussion is not intended for the authors. Not sure if you already knew it but I learned only recently from another reviewing process that you can revise the draft. Since the change I expect is relatively significant, I hope you could actually do the revision to make it more clear and convince me of a recommendation. Unfortunately I am not sure if you only have two days left to do it. I will also message AC to see if they could help.
> >
> > Now let me explain what change I expect. Your reply is indeed helpful. With some additional help I realize that SVT need not refresh the threshold noise in each step. To be more concrete, I assume you refer to Algorithm 1 (correlated threshold) instead of Algorithm 2 (refreshed threshold) [here](https://arxiv.org/pdf/1603.01699.pdf). Alg 1 is indeed not known to be a composition AFAIK, while Alg 2 (which is from an arguably more standard and better-known source) is analyzed exactly via traditional composition.
> >
> > In addition, using correlated thresholds is important because according to the same source
> > >... [refreshing] causes Alg. 2 to have significantly worse performance than Alg. 1, as we show in Section 6
> > The wastefulness is more significant in composition, according to e.g. Theorem 11 of [this paper](https://proceedings.neurips.cc/paper/2020/file/e9bf14a419d77534105016f5ec122d62-Paper.pdf)
> >
> > If you intend to use this as an example, I would like you to be very explicit in the paper about these points, which in my opinion are better off not regarded as common knowledge. If you have a less tricky example, please go ahead with it.

---

> > > ### Author Response · Authors · 2022-08-07
> > > **We will revise the draft & additional comments**
> > >
> > > Dear Reviewer 2XK9,
> > >
> > > Thanks for your updates. We are happy that you agree with a part of our arguments. We will try our best to update the current draft. Meanwhile, let us further clarify several points of our contribution:
> > >
> > > (1) You mentioned that one can use our composition theorem to compose Alg 1 in [(Lyu et al.)](https://arxiv.org/pdf/1603.01699.pdf) with other interactive mechanisms, which was not known before. Thank you for pointing this out. However, we would like to stress that our composition theorem also says something non-trivial even when one composes Alg 2 in [(Lyu et al.)](https://arxiv.org/pdf/1603.01699.pdf) with other mechanisms.
> > >
> > > Consider running the standard AboveThreshold algorithm (i.e., Alg 2), which we denote as algorithm A in the following. Under the traditional analysis, the user **has to** keep issuing queries to A until getting one "AboveThreshold" outcome. However, it might be possible that when the user gets several "BelowThreshold" results from A, he/she decides to query another concurrently running mechanism B for a while (note that we **don't** refresh the threshold for A at this moment, since we haven't received a positive outcome yet), and then switches back to query A again. It is unclear how to deal with this scenario using the standard composition theorem, because the second list of queries to A depends on the result of B, which in turn depends on the private information. In contrast, using our concurrent composition theorem reveals that the overall privacy loss is the combination of the loss of A and B as if one were running two mechanisms sequentially. The saving might not be significant when we compose two mechanisms. However, when we compose k concurrently running mechanisms, the privacy loss scales only proportionally with $\sqrt{k}$, which matches the privacy loss of sequential compositions, and is arguably appealing.
> > >
> > > (2) Besides the theoretical interests, our theorem has implications for practical deployments of interactive DP mechanisms. Suppose, for example, a data center holds the private information of individuals and offers data analysts access to the database (interactively and differentially-privately). Without knowing the concurrent composition theorem, it might be possible that k analysts collude by coordinating their queries to the database and extracting much more sensitive information. Our result strongly refutes the possibility of such an attack. In particular, Suppose every data analyst has only $\varepsilon$-privacy budget. Then, even if they collude and spend their privacy budget in whatever way, their computation result is still $\approx O(\sqrt{k} \varepsilon)$-DP w.r.t. the private database.
> > >
> > > Thank you again for your suggestions. We will revise the current manuscript to the best possible extent before the discussion deadline.

---

> > > > ### Comment · Reviewer_2XK9 · 2022-08-08
> > > > **reply**
> > > >
> > > > Thanks for the prompt reply. In this example, can't I just modify the mechanism so that whenever a switch happens, the noise must be refreshed when it comes back? Then I assume traditional composition still applies. How much improvement does concurrent composition incur?
> > > >
> > > > In addition, none of these examples seems to involve $\delta$. Is it really important to have a concurrent composition theorem with $\delta$?
> > > >
> > > > I feel sorry to ask you to do the job of [Vadhan-Wang'21], but please include these examples and work them out. Despite the question above, I would prefer an example as simple as possible (i.e. possibly without $\delta$). Even if the example justifies the conceptual significance of [Vadhan-Wang'21] more than this work, it would be a good deal of contribution and help readers like me.

---

> > > > > ### Author Response · Authors · 2022-08-08
> > > > > **About the switching cost of SVT**
> > > > >
> > > > > Thank you for the question: We can refresh the noise whenever a switching happens. However, each time we refresh the noise, we have halted the current execution of SVT and started a new one. Then, if the switching happens for $T$ times, we need to compose $T$ executions of the SVT algorithm sequentially (under the traditional composition). In this case, the privacy loss scales proportionally with $\sqrt{T}$, which can be very large. In contrast, using the concurrent composition theorem reveals that the privacy loss scales proportionally with $\sqrt{c}$, where $c$ is the number of "AboveThreshold" results.
> > > > >
> > > > > Regarding the case of $\delta > 0$, a good example might be the private multiplicative update algorithm by Hardt and Rothblum (see, e.g. https://guyrothblum.files.wordpress.com/2014/11/hr10.pdf). The HR algorithm promises to answer $m$ adaptive linear queries while satisfying $(\varepsilon, \delta)$-DP for some non-zero $\delta$. If one were to run the HR algorithm concurrently with other private mechanisms, it seems hopeless to get a reasonable privacy guarantee under traditional composition. In particular, it seems difficult to break the HR analysis into $m$ individual pieces for each query.
> > > > >
> > > > > We would be happy to include a section of motivating examples if it can better motivate readers!

---

> > > > > ### Author Response · Authors · 2022-08-08
> > > > > **We have uploaded a revision**
> > > > >
> > > > > Hi, Reviewer 2XK9,
> > > > >
> > > > > We are happy to let you know that we have uploaded a revision of our manuscript. In this revision, you can find in Section 3 a high-level discussion about the significance of the concurrent composition framework. In Appendix B of the supplementary material, you can find an example algorithm and its (surprisingly simple) analysis under the concurrent composition framework.
> > > > >
> > > > > Please let us know if you have any further questions.

---

### Official Review · Reviewer_wWSd · 2022-07-21

**Rating:** 5
**Confidence:** 5
**Soundness:** 3 good
**Presentation:** 3 good
**Contribution:** 3 good

**Summary:**

This paper studies the concurrent composition of interactive differentially private mechanisms. It shows optimal parallel composition properties of approximate DP and RDP.

**Questions:**

See the weaknesses.

**Limitations:**

Yes.

**Strengths And Weaknesses:**

Strengths: This paper studies an important problem. Currently, interactivity is one of the major obstacles in DP deployment. The previous optimal composition theorems only hold for sequential composition. Studying concurrent composition is timely research.

Weaknesses: Most of the paper is sound to me, however, I have a few concerns.

1. My main concern 1: on page 2, in the setup, the paper describes that ``A first computes a pair ... sends a query x_1 to M_{i_1} and gets the response.'', which is only one case in the interactive setting. The algorithm can send the first message to adversary A as well. The proof technique for the two cases might be different. For example, it claims that ``Again, we modify A to a new mechanism A' first queries RR...'' on page 5, which is only for one case.

2. My main concern 2: I'm not sure if Corollary 1 is immediate from Theorem 2, since zCDP allows for varying \alpha in the Renyi divergence. I don't see how the proof can be directly generalized here. This paper needs to justify this claim.

3. In the proof of RDP, the proof assumes the order of the messages, which needs justification. The order lemma in [VW21] only holds for approximate DP but not RDP, although I'm convinced it's true (still, there should be a lemma to prove this before using this fact.)

---

> ### Author Response · Authors · 2022-07-28
> **Responses to your concerns**
>
> We thank Reviewer wWSd for the review and questions. Here is our response to your concerns:
>
> 1. Yes, we assume that it is always the adversary (instead of algorithms) who sends the first message.
>    This is without loss of generality: we can add a special "initialization" query to the query domain. Then, we require that every valid adversary must send the "initialization" query to the algorithm in the first round. Having received the "initialization" query, the algorithm can respond with either a "SUCCESS" symbol (if they do not have a message to say), or whatever message they want to send.
>
> 2. Assuming Theorem 2, we prove Corollary 1 here: Suppose M1 and M2 are two mechanisms satisfying $r_1$-zCDP and $r_2$-zCDP, respectively. By definition, this is to say that they satisfy $(\alpha,\alpha\cdot r_1)$ (resp. $(\alpha,\alpha \cdot r_2)$)-RDP for every $\alpha > 1$. By our composition theorem, this implies that for every $\alpha > 1$, $COMP(M_1,M_2)$ satisfies $(\alpha, \alpha(r_1+r_2))$-zCDP, which, by definition, implies that $COMP(M_1,M_2)$ satisfies $(r_1+r_2)$-zCDP. The proof for truncated CDP is similar, except that we will argue for a range of $\alpha \in (1,\omega)$ instead of $\alpha \in (1,\infty)$.
> 3. In Line 311-314 of the submission, we mentioned a simple reduction to justify why we can assume that the order of messages is fixed. Let us re-state the reduction here for convenience: if the current mechanism is not the one that the adversary wants to speak with, the adversary can send a special "SKIP" query. The current mechanism then responds with a fixed message. This does not leak any information and allows the adversary to switch to the next mechanism. We will add a lemma to formally implement this reduction in the final version of the paper.
>
> We hope the response above answers your questions. In summary, we believe all of your concerns can be clarified with relatively quick proofs. Thank you again for pointing out your concerns: fixing them would definitely make our paper more clear and accessible. Please let us know if you have further questions.

---

> > ### Comment · Reviewer_wWSd · 2022-08-09
> > **Response**
> >
> > Thank you for your response. I have no additional comments.

---

> > > ### Author Response · Authors · 2022-08-09
> > > **Thank you**
> > >
> > > Thank you Reviewer wWSd for your time and effort in reviewing our manuscript and participating in the discussion! We hope our responses were clear and convincing. Meanwhile, we would like to gently ask you to re-evaluate the significance of our work: any critiques are also welcome!
> > >
> > > Thanks,
> > > The Authors.

---

> > > > ### Comment · Reviewer_wWSd · 2022-08-09
> > > > **Response**
> > > >
> > > > I raised my score and hope you can include these formal justifications in the final version. I have an additional question, will assuming the adversary sending the first query affect the construction of \mathcal{E}^0/1? Specifically, it's constructed by rounds, and also Claim 2 is proved by backward induction.

---

> > > > > ### Author Response · Authors · 2022-08-09
> > > > > **Response**
> > > > >
> > > > > No, our proof of Claim 2 (and all other lemmas for approximate DP) were proved under the assumption that the adversary sends the first message. What we will do is we will add a reduction to justify this assumption. After adding the new reduction, we don't need to change the current proof structure.
> > > > >
> > > > > Thank you for your support. We are happy to answer any further questions you might have.

---

> > > > > > ### Comment · Reviewer_wWSd · 2022-08-10
> > > > > > **response**
> > > > > >
> > > > > > Somehow I don't think only adding a reduction to the case that the adversary sends the first message would be enough. Maybe I'm wrong, I feel the logic is the other way around. Like because the induction in your proof is applying to a round that consists of a query and an answer, so we can assume that the adversary always sends the first message. What do you think?

---

> > > > > > > ### Author Response · Authors · 2022-08-10
> > > > > > > **response**
> > > > > > >
> > > > > > > Yes, in our proof, one round of communication consists of both a query and an answer (we will make this point clear in the revision), and our proof can handle any finite rounds of communication. So, in each round, we can safely assume that the adversary speaks first (if the adversary doesn't have anything to say in this round, they just send an empty message).
> > > > > > >
> > > > > > > We thought your question was on the case that the algorithm can "start" the communication by sending the first message. To address this case, we reduce it to a new communication protocol where the adversary first sends an "initialization" message to the algorithm. The algorithm then responds with the start message it wanted to send in the original protocol.

---

### Author Response · Authors · 2022-08-08
**A Revision of the Manuscript**

Dear Reviewers,

We have uploaded a revision of our manuscript (both the main submission and the supplementary material (i.e. proofs)). It is absolutely OK if you don't have time to read the new revision. For your convenience, we would like to quickly highlight the content of the revision:

1. After the submission deadline, we were made aware of this independent and concurrent work (https://arxiv.org/abs/2207.08335). This revision mentions this concurrent work and adds a brief comparison with their results (Lines 153 - 160).

2. As suggested by Reviewer 2XK9, we added a section in the main submission (the new Section 3, Lines 161 - 196) to demonstrate the implications of concurrent composition theorems. We also included an example algorithm and its analysis in Appendix B, which can be found in the supplementary material.

3. Other changes: e.g. we added several brief justifications for concerns raised by Reviewer wWSd, improved some texts, and fixed many typos.

Due to the time limit and the current 9-page space limit, we haven't addressed every comment perfectly. We will continue to improve the write-up. You are also more than welcome to add any additional comments.

Thanks,
Paper7102 Authors

---

### Meta-Review · Area_Chair_vrLh · 2022-08-24

**Recommendation:** Accept
**Confidence:** Certain

**Metareview:**

The paper proposes proves optimal parallel composition theorems for major notions of differential privacy (approximate DP, Renyi DP, and zero-concentrated DP). We believe that this is an interesting theoretical contribution to the DP literature. We encourage the authors to incorporate the comments from the reviewers to make it more interesting for the ML audience.

**Award:**

No

---

### Decision · Program_Chairs · 2022-09-14

Accept